# Upper Confidence Primal-Dual Reinforcement Learning for CMDP with Adversarial Loss

**Shuang Qiu**[1]    **Xiaohan Wei**[2]    **Zhuoran Yang**[3]    **Jieping Ye**[1,4]    **Zhaoran Wang**[5]

[1]University of Michigan    [2]Facebook, Inc.    [3]Princeton University
[4]AI Lab, Didi Chuxing    [5]Northwestern University

qiush@umich.edu    ubimeteor@fb.com    zy6@princeton.edu
jpye@umich.edu    zhaoranwang@gmail.com

## Abstract

We consider online learning for episodic stochastically constrained Markov decision processes (CMDP), which plays a central role in ensuring the safety of reinforcement learning. Here the loss function can vary arbitrarily across the episodes, and both the loss received and the budget consumption are revealed at the end of each episode. Previous works solve this problem under the restrictive assumption that the transition model of the MDP is known a priori and establish regret bounds that depend polynomially on the cardinalities of the state space $\mathcal{S}$ and the action space $\mathcal{A}$. In this work, we propose a new *upper confidence primal-dual* algorithm, which only requires the trajectories sampled from the transition model. In particular, we prove that the proposed algorithm achieves $\widetilde{\mathcal{O}}(L|\mathcal{S}|\sqrt{|\mathcal{A}|T})$ upper bounds of both the regret and the constraint violation, where $L$ is the length of each episode. Our analysis incorporates a new high-probability drift analysis of Lagrange multiplier processes into the celebrated regret analysis of upper confidence reinforcement learning, which demonstrates the power of "optimism in the face of uncertainty" in constrained online learning.

## 1 Introduction

Constrained Markov decision processes (CMDPs) play an important role in control and planning. It aims at maximizing a reward or minimizing a penalty metric over the set of all available policies subject to constraints on other relevant metrics. The constraints aim at enforcing the fairness or safety of the policies so that overtime the behaviors of the chosen policy is under control. For example, in an edge cloud serving network [Urgaonkar et al., 2015, Wang et al., 2015], one would like to minimize the average cost of serving the moving targets subject to a constraint on the average serving delay. In an autonomous vehicle control problem [Le et al., 2019], one might be interested in minimizing the driving time subject to certain fuel efficiency or driving safety constraints.

Classical treatment of CMDPs dates back to Fox [1966], Altman [1999] reformulating the problem into a linear program (LP) via stationary state-action occupancy measures. However, to formulate such an LP, one requires the full knowledge of the transition model, reward, and constraint functions, and also assumes them to be fixed. Leveraging the episodic structure of a class of MDPs, Neely [2012] develops online renewal optimization which potentially allows the loss and constraint functions to be stochastically varying and unknown, while still relying on the transition model to solve the subproblem within the episode.

More recently, policy-search type algorithms have received much attention, attaining state-of-art performance in various control tasks without knowledge of the transition model, e.g., Williams [1992], Baxter and Bartlett [2000], Konda and Tsitsiklis [2000], Kakade [2002], Schulman et al. [2015], Lillicrap et al. [2015], Schulman et al. [2017], Sutton and Barto [2018], Fazel et al. [2018],

Abbasi-Yadkori et al. [2019a,b], Bhandari and Russo [2019], Cai et al. [2019], Wang et al. [2019], Liu et al. [2019], Agarwal et al. [2019]. While most of the algorithms focus on unconstrained policy optimization, there are efforts to develop policy-based methods in constrained MDPs where constraints are known with limited theoretical guarantees. The work Chow et al. [2017] develops a primal-dual type algorithm which is shown to converge to some constraint satisfying policy. The work Achiam et al. [2017] develops a trust-region type algorithm which requires solving an optimization problem with both trust region and safety constraints during each update. Generalizing ideas from the fitted-Q iteration, Le et al. [2019] develops a batch offline primal-dual type algorithm which guarantees only the time average primal-dual gap converges.

The goal of this paper is to solve constrained episodic MDPs with more generality in that not only are transition models unknown, but also the loss and constraint functions can change online. In particular, the losses can be arbitrarily time-varying and adversarial. When assuming the transition model is known, Even-Dar et al. [2009] achieves $\widetilde{\mathcal{O}}(\varrho^2\sqrt{T\log|\mathcal{A}|})$ regret with $\varrho$ being the mixing time of the MDP, and the work Yu et al. [2009] achieves $\widetilde{\mathcal{O}}(T^{2/3})$ regret. These two papers consider a continuous setting that is a little different to the episodic setting that we consider in this paper. The work Zimin and Neu [2013] further studies the episodic MDP and achieves $\widetilde{\mathcal{O}}(L\sqrt{T\log(|\mathcal{S}||\mathcal{A}|)})$ regret. For the constrained case with known transitions, the work Wei et al. [2018] achieves $\widetilde{\mathcal{O}}(\text{poly}(|\mathcal{S}||\mathcal{A}|)\sqrt{T})$ regret and constraint violations, and the work Zheng and Ratliff [2020] attains $\widetilde{\mathcal{O}}(|\mathcal{S}||\mathcal{A}|T^{3/4})$.

After we finished the first version of this work, there are several concurrent works appearing which also focus on CMDPs with unknown transitions and rewards. The work Efroni et al. [2020] studies episodic tabular MDPs with unknown but fixed reward and constraint functions. Leveraging upper confidence bound (UCB) on the reward, constraints and transitions, they obtain an $\mathcal{O}(\sqrt{T})$ regret and constraint violation via linear program as well as primal-dual optimization. In another work, Ding et al. [2020] studies the constrained episodic MDPs with a linear structure and adversarial losses via a primal-dual-type policy optimization algorithm, achieving $\widetilde{\mathcal{O}}(\sqrt{T})$ regret and constraint violation. While their scenario is more general than ours, their dependencies on $|\mathcal{S}|$, $|\mathcal{A}|$, $L$ is considerably worse when applied to the tabular case. Both of these two works rely on Slater condition which is also more restrictive than that of this work.

On the other hand, for unconstrained online MDPs, the idea of UCB has shown to be effective and helped achieving tight regret bounds without knowing the transition model, e.g., Jaksch et al. [2010], Azar et al. [2017], Rosenberg and Mansour [2019a,b], Jin et al. [2019]. The main idea there is to sequentially refine a confidence set of the transition model and choose a model in the interval which performs the best in optimizing the current value.

The main contribution of this paper is to show that UCB is also effective when incorporating with primal-dual type approaches to achieve $\widetilde{\mathcal{O}}(L|\mathcal{S}|\sqrt{|\mathcal{A}|T})$ regret and constraint violation simultaneously in online MDPs with no knowledge on the transition models, the loss is adversarial and the constraints are stochastic. This almost matches the lower bound $\Omega(\sqrt{L|\mathcal{S}||\mathcal{A}|T})$ for the regret Jaksch et al. [2010] up to an $\mathcal{O}(\sqrt{L|\mathcal{S}|})$ factor. Under the hood is a new Lagrange multiplier condition together with a new drift analysis on the Lagrange multipliers leading to low constraint violation. Our setup is challenging compared to classical constrained optimization in particular due to **(1)** the unknown loss and constraint functions from the online setup; **(2)** the time varying decision sets resulting from moving confidence interval estimation of UCB. The decision sets can potentially be much larger than or even inconsistent with the true decision set knowing the model, resulting in potentially large constraint violation. The main idea is to utilize a Lagrange multiplier condition as well as a confidence bound of the model to construct a probabilistic bound on an online dual multiplier. We then explicitly take into account the laziness nature of the UCB estimation in our algorithm to argue that the bound on the dual multiplier gives the bound on constraint violation.

## 2 Problem Formulation

Consider an episodic loop-free MDP with a finite state space $\mathcal{S}$ and a finite action space $\mathcal{A}$ at each state over a finite horizon of $T$ episodes. Each episode starts with a fixed initial state $s_0$ and ends with a terminal state $s_L$. The transition probability is $P : \mathcal{S} \times \mathcal{S} \times \mathcal{A} \mapsto [0,1]$, where $P(s'|s,a)$ gives the probability of transition from $s$ to $s'$ under an action $a$. This underlying transition model $P$ is assumed to be *unknown*. The state space is divided into layers with a loop-free structure, i.e.,

$\mathcal{S} := \mathcal{S}_0 \cup \mathcal{S}_1 \cup \cdots \cup \mathcal{S}_L$ with a singleton initial layer $\mathcal{S}_0 = \{s_0\}$ and terminal layer $S_L = \{s_L\}$. Furthermore, we have $\mathcal{S}_k \cap \mathcal{S}_\ell = \emptyset$ for $k \neq \ell$, and transitions are only allowed between consecutive layers, which is $P(s'|s, a) > 0$ only if $s' \in \mathcal{S}_{k+1}$, $s \in \mathcal{S}_k$, and $a \in \mathcal{A}$, $\forall k \in \{0, 1, \ldots, L-1\}$. Such an assumption enforces that each path from the initial state to the terminal state takes a fixed length $L$. This is not an excessively restrictive assumption as any loop-free MDP with bounded varying path lengths can be transformed into one with a fixed path length (see György et al. [2007] for details).

The loss function for each episode is $f^t : \mathcal{S} \times \mathcal{A} \times \mathcal{S} \mapsto \mathbb{R}$, where $f^t(s, a, s')$ denotes the loss received at episode $t$ for any $s \in \mathcal{S}_k$, $s' \in \mathcal{S}_{k+1}$, and $a \in \mathcal{A}$, $\forall k \in \{0, 1, 2, \ldots, L-1\}$. We assume $f_t$ can be arbitrarily varying with potentially no fixed probability distribution. There are $I$ stochastic constraint (or budget consumption) functions: $g_i^t : \mathcal{S} \times \mathcal{A} \times \mathcal{S} \mapsto \mathbb{R}$, $\forall i \in \{1, 2, \ldots, I\}$, where $g_i^t(s, a, s')$ denotes the price to pay at episode $t$ for any $(s, a, s')$. Each stochastic function $g_i^t$ at episode $t$ is sampled according to a random variable $\xi_i^t \sim \mathcal{D}_i$, namely $g_i^t(s, a, s') = g_i(s, a, s'; \xi_i^t)$. Then, we define $g_i(s, a, s') := \mathbb{E}[g_i^t(s, a, s')] = \mathbb{E}[g_i(s, a, s'; \xi_i^t)]$ where the expectation is taken over the randomness of $\xi_i^t \sim \mathcal{D}_i$. For abbreviation, we denote $g_i = \mathbb{E}[g_i^t]$. In addition, the functions $f^t$ and $g_i^t$, $\forall i \in \{1, \ldots, I\}$, are mutually independent and independent of the Markov transition. Both the loss functions and the budget consumption functions are revealed at the end of each episode.

**Remark 2.1.** *It might be tempting to consider the more general scenario that both losses and constraints are arbitrarily time varying. For such a setting, however, there exist counterexamples [Mannor et al., 2009] in the arguably simpler constrained online learning scenario that no algorithm can achieve sublinear regret and constraint violation simultaneously. Therefore, we seek to put extra assumptions on the problem so that obtaining sublinear regret and constraint violation is feasible, one of which is to assert constraints to be stochastic instead of arbitrarily varying.*

For any episode $t$, a policy $\pi_t$ is the conditional probability $\pi_t(a|s)$ of choosing an action $a \in \mathcal{A}$ at any given state $s \in \mathcal{S}$. Let $(s_k, a_k, s_{k+1}) \in \mathcal{S}_k \times \mathcal{A} \times \mathcal{S}_{k+1}$ denotes a random tuple generated according to the transition model $P$ and the policy $\pi_t$. The corresponding expected loss is $\mathbb{E}[\sum_{k=0}^{L-1} f^t(s_k, a_k, s_{k+1})|\pi_t, P]$, while the budget costs are $\mathbb{E}[\sum_{k=0}^{L-1} g_i^t(s_k, a_k, s_{k+1})|\pi_t, P], i \in \{1, \cdots, I\}$, where the expectations are taken w.r.t. the randomness of the tuples $(s_k, a_k, s_{k+1})$.

In this paper, we adopt the occupancy measure $\theta(s, a, s')$ for our analysis. In general, the occupancy measure $\theta(s, a, s')$ is a joint probability of the tuple $(s, a, s') \in \mathcal{S} \times \mathcal{A} \times \mathcal{S}$ under some certain policy and transition model. Particularly, with the true transition $P$, we define the set as

$$\Delta = \{\theta \; : \; \theta \text{ satisfies the conditions (a), (b), and (c)}\},$$

where the conditions (a) (b) (c) [Altman, 1999] are

(a) $\sum_{s \in \mathcal{S}_k} \sum_{a \in \mathcal{A}} \sum_{s' \in \mathcal{S}_{k+1}} \theta(s, a, s') = 1, \forall k \in \{0, \ldots, L-1\}$, and $\theta(s, a, s') \geq 0$.
(b) $\sum_{s \in \mathcal{S}_k} \sum_{a \in \mathcal{A}} \theta(s, a, s') = \sum_{a \in \mathcal{A}} \sum_{s'' \in \mathcal{S}_{k+2}} \theta(s', a, s''), \forall k \in \{0, \ldots, L-2\}, s' \in \mathcal{S}_{k+1}$.
(c) $\frac{\theta(s, a, s')}{\sum_{s'' \in \mathcal{S}_{k+1}} \theta(s, a, s'')} = P(s'|s, a), \forall k \in \{0, \ldots, L-1\}, s \in \mathcal{S}_k, a \in \mathcal{A}, s' \in \mathcal{S}_{k+1}$.

We can further recover a policy $\pi$ from $\theta$ via $\pi(a|s) = \frac{\sum_{s' \in \mathcal{S}_{k+1}} \theta(s, a, s')}{\sum_{s' \in \mathcal{S}_{k+1}, a \in \mathcal{A}} \theta(s, a, s')}$ for any $(s, a) \in \mathcal{S}_k \times \mathcal{A}$.

We define $\overline{\theta}^t(s, a, s')$ to be the occupancy measure at episode $t$ w.r.t. the true transition $P$, resulting from a policy $\pi_t$ at episode $t$. Given the definition of occupancy measure, we can rewrite the expected loss and the budget cost as $\mathbb{E}[\sum_{k=0}^{L-1} f^t(s_k, a_k, s_{k+1})|\pi_t, P] = \langle f^t, \overline{\theta}^t \rangle$ where $\langle f^t, \overline{\theta}^t \rangle = \sum_{s,a,s'} f^t(s, a, s')\overline{\theta}^t(s, a, s')$ and $\mathbb{E}[\sum_{k=0}^{L-1} g_i^t(s_k, a_k, s_{k+1})|\pi_t, P] = \langle g_i^t, \overline{\theta}^t \rangle$ with $\langle g_i^t, \overline{\theta}^t \rangle = \sum_{s,a,s'} f^t(s, a, s')\overline{\theta}^t(s, a, s')$. We aim to solve the following constrained optimization, and let $\overline{\theta}^*$ be one solution which is further viewed as a reference point to define the regret:

$$\underset{\theta \in \Delta}{\text{minimize}} \quad \sum_{t=0}^{T-1} \langle f^t, \theta \rangle, \quad \text{subject to} \quad \langle g_i, \theta \rangle \leq c_i, \quad \forall i \in \{1, 2, \ldots, I\}, \tag{1}$$

where $\sum_{t=0}^{T-1} \langle f^t, \theta \rangle = \sum_{t=0}^{T-1} \mathbb{E}[\sum_{k=0}^{L-1} f^t(s_k, a_k, s_{k+1})|\pi, P]$ is the overall loss in $T$ episodes and constraints are enforced on the budget cost $\langle g_i, \theta \rangle = \mathbb{E}[\sum_{k=0}^{L-1} g_i(s_k, a_k, s_{k+1})|\pi, P]$ based on the expected budget consumption functions $g_i, \forall i \in [I]$. To measure the regret and the constraint violation

respectively for solving the above problem in an online setting , we define the following two metrics:

$$\text{Regret}(T) := \sum_{t=0}^{T-1} \langle f^t, \overline{\theta}^t - \overline{\theta}^* \rangle, \quad \text{and} \quad \text{Violation}(T) := \left\| \left[ \sum_{t=0}^{T-1} \left( \mathbf{g}(\overline{\theta}^t) - \mathbf{c} \right) \right]_+ \right\|_2, \quad (2)$$

where the notation $[\mathbf{v}]_+$ denotes the entry-wise application of $\max\{\cdot, 0\}$ for any vector $\mathbf{v}$. For abbreviation, we let $\mathbf{g}^t(\theta) := [\langle g_1^t, \theta \rangle, \cdots, \langle g_I^t, \theta \rangle]^\top$, and $\mathbf{c} := [c_1, \cdots, c_I]^\top$.

The goal is to attain a sublinear regret bound and constraint violation on this problem w.r.t. *any fixed stationary policy $\pi$, which does not change over episodes*. In another word, we compare to the best policy $\pi^*$ in hindsight whose corresponding occupancy measure $\overline{\theta}^* \in \Delta$ solves problem (1). We make the following assumption on the existence of a solution to (1).

**Assumption 2.2.** *There exists at least one fixed policy $\pi$ such that the corresponding occupancy measure $\theta \in \Delta$ is feasible, i.e., $\langle g_i, \theta \rangle \le c_i, \forall i \in \{1, 2, \cdots, I\}$.*

Then, we assume boundedness on function values for simplicity of notations without loss of generality.

**Assumption 2.3.** *We assume the following quantities are bounded. For any $t \in \{1, 2, \ldots, T\}$, (1) $\sup_{s,a,s'} |f^t(s, a, s')| \le 1$, (2) $\sum_{i=1}^{I} \sup_{s,a,s'} |g_i^t(s, a, s')| \le 1$, (3) $\sum_{i=1}^{I} |c_i| \le L$.*

When the transition model $P$ is known and Slater's condition holds (i.e., existence of a policy which satisfies all stochastic inequality constraints with a constant $\varepsilon$-slackness), this stochastically constrained online linear program can be solved via similar methods as Wei et al. [2018], Yu et al. [2017] with a regret bound that depends polynomially on the cardinalities of state and action spaces, which is highly suboptimal especially when the state or action space is large. The main challenge we will address in this paper is to *solve this problem without knowing the model $P$, or losses and constraints before making decisions, while tightening the dependency on both state and action spaces in the resulting performance bound.*

## 3 Proposed Algorithm

In this section, we introduce our proposed algorithm, namely, the upper confidence primal-dual (UCPD) algorithm, as presented in Algorithm 1. It adopts a primal-dual mirror descent type algorithm solving constrained problems but with an important difference: we maintain a confidence set via past sample trajectories, which contains the true MDP model $P$ with high probability, and choose the policy to minimize the proximal Lagrangian using the most optimistic model from the confidence set. Such an idea, known as *optimism in the face of uncertainty*, is reminiscent of the upper confidence bound (UCB) algorithm [Auer et al., 2002] for stochastic multi-armed bandit (MAB) and first proposed by Jaksch et al. [2010] to obtain a near-optimal regret for reinforcement learning problems.

In the algorithm, we introduce epochs, which are back-to-back time intervals that span several episodes. We use $\ell \in \{1, 2, \cdots\}$ to index the epochs and use $\ell(t)$ to denote a mapping from the episode index $t$ to the epoch index, indicating which epoch the $t$-th episode lives. Next, let $N_\ell(s, a)$ and $M_\ell(s, a, s')$ be two global counters which indicate the number of times the tuples $(s, a)$ and $(s, a, s')$ appear before the $\ell$-th epoch. Let $n_\ell(s, a), m_\ell(s, a, s')$ be two local counters which indicate the number of times the tuples $(s, a)$ and $(s, a, s')$ appear in the $\ell$-th epoch. We start a new epoch whenever there exists $(s, a)$ such that $n_{\ell(t)}(s, a) \ge N_{\ell(t)}(s, a)$. Otherwise, set $\ell(t + 1) = \ell(t)$. Such an update rule follows from Jaksch et al. [2010]. Then, we define the empirical transition model $\widehat{P}_\ell$ at any epoch $\ell > 0$ as $\widehat{P}_\ell(s'|s, a) := M_\ell(s, a, s') / \max\{1, N_\ell(s, a)\}, \quad \forall s, s' \in \mathcal{S}, a \in \mathcal{A}$. As shown in Remark 5.8, introducing the 'epoch' is necessary to achieve an $\widetilde{\mathcal{O}}(\sqrt{T})$ constraint violation.

The next lemma shows that with high probability, the true transition model $P$ is contained in a confidence interval around the empirical one no matter what sequence of policies taken.

**Lemma 3.1** (Lemma 1 of Neu et al. [2012])**.** *For any $\zeta \in (0, 1)$, we have that with probability at least $1 - \zeta$, for all epoch $\ell$ and any state and action pair $(s, a) \in \mathcal{S} \times \mathcal{A}$, $\|P(\cdot|s, a) - \widehat{P}_\ell(\cdot|s, a)\|_1 \le \varepsilon_\ell^\zeta(s, a)$, with the error $\varepsilon_\ell^\zeta(s, a)$ being*

$$\varepsilon_\ell^\zeta(s, a) := \sqrt{\frac{2|\mathcal{S}_{k(s)+1}| \log(T|\mathcal{S}||\mathcal{A}|/\zeta)}{\max\{1, N_\ell(s, a)\}}}, \quad (3)$$

*where $k(s)$ is a mapping from state $s$ to the layer index $k$, indicating whe layer the state $s$ belongs to.*

---
**Algorithm 1** Upper-Confidence Primal-Dual (UCPD) Mirror Descent
---
1: **Input:** Let $V, \alpha > 0, \lambda \in [0, 1)$ be some trade-off parameters. Fix $\zeta \in (0, 1)$.
2: **Initialize:** $Q_i(0) = 0$, $\forall i = 1, \ldots, I$. $\theta^0(s, a, s') = 1/(|\mathcal{S}_k||\mathcal{S}_{k+1}||\mathcal{A}|)$, $\forall (s, a, s') \in \mathcal{S}_k \times \mathcal{A} \times \mathcal{S}_{k+1}$. $\ell(1) = 1$. $n_1(s, a) = 0$, $N_1(s, a) = 0$, $\forall (s, a) \in \mathcal{S} \times \mathcal{A}$. $m_1(s, a, s') = 0$, $M_1(s, a, s') = 0$, $f^0(s, a, s') = 0$, $g^0(s, a, s') = 0$, $\widehat{P}_1(s'|s, a) = 0$, $\forall (s, a, s') \in \mathcal{S} \times \mathcal{A} \times \mathcal{S}$.
3: **for** $t = 1, \ldots, T$ **do**
4:    Compute $\theta^t$ via (6) and the corresponding policy $\pi_t$ via (4).
5:    Sample a path $(s_0^t, a_0^t, \cdots, s_{L-1}^t, a_{L-1}^t, s_L^t)$ following the policy $\pi_t$.
6:    Update each dual multiplier $Q_i(t)$ via (5) and update the local counters:
$$n_{\ell(t)}(s_k^t, a_k^t) = n_{\ell(t)}(s_k^t, a_k^t) + 1, \quad m_{\ell(t)}(s_k^t, a_k^t, s_{k+1}^t) = m_{\ell(t)}(s_k^t, a_k^t, s_{k+1}^t) + 1.$$
7:    Observe the loss function $f^t$ and constraint functions $\{g_i^t\}_{i=1}^I$.
8:    **if** $\exists (s, a) \in \mathcal{S} \times \mathcal{A}$, $n_{\ell(t)}(s, a) \geq N_{\ell(t)}(s, a)$, **then**
9:       **Start a new epoch:**
10:      Set $\ell(t + 1) = \ell(t) + 1$, and update the global counters for all $s, s' \in \mathcal{S}$, $a \in \mathcal{A}$ by
$$N_{\ell(t+1)}(s, a) = N_{\ell(t)}(s, a) + n_{\ell(t)}(s, a),$$
$$M_{\ell(t+1)}(s, a, s') = M_{\ell(t)}(s, a, s') + m_{\ell(t)}(s, a, s').$$
11:      Construct the empirical transition $\widehat{P}_{\ell(t+1)}(s'|s, a) := \frac{M_{\ell(t+1)}(s,a,s')}{\max\{1, N_{\ell(t+1)}(s,a)\}}, \forall (s, a, s')$.
12:      Initialize $n_{\ell(t+1)}(s, a) = 0$, $m_{\ell(t+1)}(s, a, s') = 0$, $\forall (s, a, s') \in \mathcal{S} \times \mathcal{A} \times \mathcal{S}$.
13:   **else**
14:      Set $\ell(t + 1) = \ell(t)$.
15:   **end if**
16: **end for**
---

## 3.1 Computing Optimistic Policies

Next, we show how to compute the policy at each episode. Formally, we introduce a new occupancy measure at episode $t$, namely $\theta^t(s, a, s')$, $s, s' \in \mathcal{S}$, $a \in \mathcal{A}$. It should be emphasized that this is different from the $\overline{\theta}^t(s, a, s')$ defined in the previous section as $\theta^t(s, a, s')$ is chosen by the decision maker at episode $t$ to construct the policy. In particular, $\theta^t(s, a, s')$ does not have to satisfy the local balance equation (c). Once getting $\theta^t(s, a, s')$ (which will be detailed below), we construct the policy:

$$\pi_t(a|s) = \frac{\sum_{s'} \theta^t(s, a, s')}{\sum_{s', a} \theta^t(s, a, s')}, \ \forall a \in \mathcal{A}, \ s \in \mathcal{S}. \tag{4}$$

Next, we demonstrate the proposed method computing $\theta^t(s, a, s')$. First, we introduce an online dual multiplier $Q_i(t)$ for each constraint in (1), which is 0 when $t = 0$ and updated as follows for $t \geq 1$

$$Q_i(t) = \max\{Q_i(t-1) + \langle g_i^{t-1}, \theta^t \rangle - c_i, \ 0\}. \tag{5}$$

At each episode, we compute the occupancy measure $\theta^t(s, a, s')$ by solving an optimistic regularized linear program (ORLP) with tuning parameters $\lambda, V, \alpha > 0$. Specifically, we update $\theta^t$ by

$$\theta^t = \operatorname*{argmin}_{\theta \in \Delta(\ell(t), \zeta)} \ \left\langle V f^{t-1} + \sum_{i=1}^I Q_i(t-1) g_i^{t-1}, \theta \right\rangle + \alpha D(\theta, \widetilde{\theta}^{t-1}), \tag{6}$$

which introduces extra notations $\Delta(\ell(t), \zeta)$, $\widetilde{\theta}^{t-1}$, and $D(\cdot, \cdot)$ that will be elaborated below. Specifically, we denote by $D(\cdot, \cdot)$ the unnormalized Kullback-Leibler (KL) divergence, which is defined as $D(\theta, \theta') := \sum_{s,a,s'} [\theta(s, a, s') \log \frac{\theta(s,a,s')}{\theta'(s,a,s')} - \theta(s, a, s') + \theta'(s, a, s')]$, $\forall \theta, \theta'$. In addition, for $\forall k = \{0, \ldots, L-1\}$ and $\forall s \in \mathcal{S}_k, a \in \mathcal{A}, s' \in \mathcal{S}_{k+1}$, we compute $\widetilde{\theta}^{t-1}$ via $\widetilde{\theta}^{t-1}(s, a, s') = (1-\lambda)\theta^{t-1}(s, a, s') + \frac{\lambda}{|\mathcal{S}_k||\mathcal{S}_{k+1}||\mathcal{A}|}$, where $0 \leq \lambda \leq 1$. This equation introduces a probability mixing, pushing the update away from the boundary and encourage explorations.

Furthermore, since for any epoch $\ell > 0$, we can compute the empirical transition model $\widehat{P}_\ell$ with the confidence interval size $\varepsilon_\ell^\zeta$ as defined in (3), we let every $\theta \in \Delta(\ell, \zeta)$ satisfy that

$$\left\| \frac{\theta(s, a, \cdot)}{\sum_{s'} \theta(s, a, s')} - \widehat{P}_\ell(\cdot|s, a) \right\|_1 \leq \varepsilon_\ell^\zeta(s, a), \ \forall s \in \mathcal{S}, a \in \mathcal{A}, \tag{7}$$

such that we can define the feasible set $\Delta(\ell, \zeta)$ for the optimization problem (6) as follows

$$\Delta(\ell, \zeta) := \{\theta : \theta \text{ satisfies conditions (a), (b), and (7) }\}. \tag{8}$$

By this definition, we know that $\theta^t \in \Delta(\ell(t), \zeta)$ at the epoch $\ell(t)$. On the other hand, according to Lemma 3.1, we have that with probability at least $1 - \zeta$, for all epoch $\ell$, $\Delta \subseteq \Delta(\ell, \zeta)$ holds. By Rosenberg and Mansour [2019a], the problem (6) is essentially a linear programming with a special structure that can be solved efficiently (see details in Section A of the supplementary material).

## 4  Main Results

Before presenting our results, we first make assumption on the existence of Lagrange multipliers. We define a partial average function starting from any time slot $t$ as $f^{(t,\tau)} := \frac{1}{\tau} \sum_{j=0}^{\tau-1} f^{t+j}$. Then, we consider the following static optimization problem (recalling $g_i := \mathbb{E}[g_i^t]$)

$$\operatorname*{minimize}_{\theta \in \Delta} \langle f^{(t,\tau)}, \theta \rangle \quad \text{s.t.} \quad \langle g_i, \theta \rangle \leq c_i, \ \forall i \in \{1, \ldots, I\}. \tag{9}$$

Denote the solution to this program as $\theta_{t,\tau}^*$. Define the Lagrangian dual function of (9) as

$$q^{(t,\tau)}(\eta) := \min_{\theta \in \Delta} \ \langle f^{(t,\tau)}, \theta \rangle + \sum_{i=1}^{I} \eta_i(\langle g_i, \theta \rangle - c_i),$$

where $\eta = [\eta_1, \ldots, \eta_I]^\top \in \mathbb{R}^I$ is a dual variable. We are ready to state our assumption.

**Assumption 4.1.** *For any time slot $t$ and any time period $\tau$, the set of primal optimal solution to (9) is non-empty. Furthermore, the set of Lagrange multipliers, which is $\mathcal{V}_{t,\tau}^* := \operatorname{argmax}_{\eta \in \mathbb{R}_+^I} q^{(t,\tau)}(\eta)$, is non-empty and bounded. Any vector in $\mathcal{V}_{t,\tau}^*$ is called a Lagrange multiplier associated with (9). Furthermore, let $B > 0$ be a constant such that for any $t \in \{1, \ldots, T\}$ and $\tau = \sqrt{T}$, the dual optimal set $\mathcal{V}_{t,\tau}^*$ defined above satisfies $\max_{\eta \in \mathcal{V}_{t,\tau}^*} \|\eta\|_2 \leq B$.*

As is discussed in Section B of the supplementary material, Assumption 4.1 proposes a weaker condition than the Slater condition commonly adopted in previous constrained online learning works. The following lemma further shows the relation between Assumption 4.1 and the dual function:

**Lemma 4.2.** *Suppose Assumption 4.1 holds, then for any $t \in \{0, \ldots, T-1\}$ and $\tau = \sqrt{T}$, there exists constants $\vartheta$, $\sigma > 0$ such that for any $\eta \in \mathbb{R}^I$ satisfying [1] $\operatorname{dist}(\eta, \mathcal{V}_{t,\tau}^*) \geq \vartheta$, we have*

$$q^{(t,\tau)}(\eta_{t,\tau}^*) - q^{(t,\tau)}(\eta) \geq \sigma \cdot \operatorname{dist}(\eta, \mathcal{V}_{t,\tau}^*), \ \ \forall \eta_{t,\tau}^* \in \mathcal{V}_{t,\tau}^*.$$

Based on the above assumptions and lemmas, we present results of the regret and constraint violation.

**Theorem 4.3.** *Consider any fixed horizon $T \geq |\mathcal{S}||\mathcal{A}|$ with $|\mathcal{S}|, |\mathcal{A}| > 1$. Suppose Assumption 2.2, 2.3, 4.1 hold and there exist absolute constants $\overline{\sigma}$ and $\overline{\vartheta}$ such that $\sigma \geq \overline{\sigma}$ and $\vartheta \leq \overline{\vartheta}$ for all $\sigma$, $\vartheta$ in Lemma 4.2 over $t = \{0, 1, \ldots, T-1\}$ and $\tau = \sqrt{T}$. If setting $\alpha = LT$, $V = L\sqrt{T}$, $\lambda = 1/T$ and $\zeta \in (0, 1/(4 + 8L/\overline{\sigma})]$ in Algorithm 1, with probability at least $1 - 4\zeta$, we have*

$$\operatorname{Regret}(T) \leq \widetilde{\mathcal{O}}\Big(L|\mathcal{S}|\sqrt{T|\mathcal{A}|}\Big), \qquad \operatorname{Violation}(T) \leq \widetilde{\mathcal{O}}\Big(L|\mathcal{S}|\sqrt{T|\mathcal{A}|}\Big),$$

*where the notation $\widetilde{O}(\cdot)$ hides the logarithmic factors $\log^{3/2}(T/\zeta)$ and $\log(T|\mathcal{S}||\mathcal{A}|/\zeta)$.*

## 5  Theoretical Analysis

### 5.1  Proof of Regret Bound

**Lemma 5.1.** *The updating rules in Algorithm 1 ensure that with probability at least $1 - 2\zeta$,*

$$\sum_{t=0}^{T-1} \left\|\theta^t - \overline{\theta}^t\right\|_1 \leq (\sqrt{2} + 1)L|\mathcal{S}|\sqrt{2T|\mathcal{A}| \ln \frac{T|\mathcal{S}||\mathcal{A}|}{\zeta}} + 2L^2\sqrt{2T \ln \frac{L}{\zeta}}.$$

**Lemma 5.2.** *The updating rules in Algorithm 1 ensure that with probability at least $1 - \zeta$,*

$$\sum_{t=0}^{T-1} \langle f^t, \theta^t - \overline{\theta}^* \rangle \leq \frac{4L^2T + (\lambda T + 1)\alpha L \log |\mathcal{S}|^2|\mathcal{A}|}{V} + 2\lambda LT + \frac{LT}{2\alpha} + \frac{1}{V} \sum_{t=0}^{T-1} \langle \mathbf{Q}(t), \mathbf{g}^t(\overline{\theta}^*) - \mathbf{c} \rangle.$$

Here we let $\mathbf{Q}(t) := [Q_1(t),\ Q_2(t),\ \cdots,\ Q_I(t)]^\top$. Next, we present Lemma 5.3, which is one of the key lemmas in our proof. Then, this lemma indicates that $\|\mathbf{Q}(t)\|_2$ is bounded by $\mathcal{O}(\sqrt{T})$ with high probability when setting the parameters $\tau, V, \alpha, \lambda$ as in Theorem 4.3. Thus, introducing stochastic constraints retains the $\mathcal{O}(\sqrt{T})$ regret. Moreover, this lemma will lead to constraint violation in the level of $\mathcal{O}(\sqrt{T})$. Lemma 5.3 is proved by making use of Assumption 4.1 and Lemma 4.2.

**Lemma 5.3.** *Letting $\tau = \sqrt{T}$ and $\zeta$ satisfy $\overline{\sigma}/4 \geq \zeta(\overline{\sigma}/2 + 2L)$, the updating rules in Algorithm 1 ensure that with probability at least $1 - T\delta$, the following inequality holds for all $t \in \{1, \ldots, T\}$,*

$$\|\mathbf{Q}(t)\|_2 \leq \omega := \psi + \tau \frac{512L^2}{\overline{\sigma}} \log \left(1 + \frac{128L^2}{\overline{\sigma}^2} e^{\overline{\sigma}/(32L)}\right) + \tau \frac{64L^2}{\overline{\sigma}} \log \frac{1}{\delta},$$

*where we define $\psi := (2\tau L + C_{V,\alpha,\lambda})/\overline{\sigma} + 2\alpha L \log(|\mathcal{S}|^2|\mathcal{A}|/\lambda)/(\overline{\sigma}\tau) + \tau\overline{\sigma}/2$ and $C_{V,\alpha,\lambda} := 2(\overline{\sigma}B + \overline{\sigma}\,\overline{\vartheta})V + (6 + 4\overline{\vartheta})VL + VL/\alpha + 4L\lambda V + 2\alpha\lambda L \log |\mathcal{S}|^2|\mathcal{A}| + 8L^2$.*

**Remark 5.4.** *We discuss the upper bound of the term $\log\left(1 + \frac{128L^2}{\overline{\sigma}^2} e^{\overline{\sigma}/(32L)}\right)$ in the following way: (1) if $\frac{128L^2}{\overline{\sigma}^2} e^{\overline{\sigma}/(32L)} \geq 1$, then this term is bounded by $\log\left(\frac{256L^2}{\overline{\sigma}^2} e^{\overline{\sigma}/(32L)}\right) = \frac{\overline{\sigma}}{32L} + \log \frac{256L^2}{\overline{\sigma}^2}$; (2) if $\frac{128L^2}{\overline{\sigma}^2} e^{\overline{\sigma}/(32L)} < 1$, then the term is bounded by $\log 2$. Thus, combining the two cases, we have $\log\left(1 + \frac{128L^2}{\overline{\sigma}^2} e^{\overline{\sigma}/(32L)}\right) \leq \log 2 + \frac{\overline{\sigma}}{32L} + \log \frac{256L^2}{\overline{\sigma}^2}$. This discussion shows that the $\log$ term in the result of Lemma 5.3 will not introduce extra dependency on $L$ except a $\log L$ term.*

With the bound of $\|\mathbf{Q}(t)\|_2$ in Lemma 5.3, we further obtain the following lemma.

**Lemma 5.5.** *By Algorithm 1, if $\overline{\sigma}/4 \geq \zeta(\overline{\sigma}/2 + 2L)$, then with probability at least $1 - 2T\delta$,*

$$\sum_{t=0}^{T-1} \langle \mathbf{Q}(t), \mathbf{g}^t(\overline{\theta}^*) - \mathbf{c} \rangle \leq 2L\omega \sqrt{T \log \frac{1}{T\delta}},$$

*with $\omega$ defined as the same as in Lemma 5.3.*

*Proof of Regret Bound in Theorem 4.3.* Recall that $\theta^t$ is the probability vector chosen by the decision maker, and $\overline{\theta}^t$ is the true occupancy measure at time $t$ while $\overline{\theta}^*$ is the solution to the problem (1). The main idea is to decompose the regret as follows

$$\sum_{t=0}^{T-1} \langle f^t, \overline{\theta}^t - \overline{\theta}^* \rangle = \sum_{t=0}^{T-1} \left( \langle f^t, \overline{\theta}^t - \theta^t \rangle + \langle f^t, \theta^t - \overline{\theta}^* \rangle \right)$$

$$\leq \underbrace{\sum_{t=0}^{T-1} \|\overline{\theta}^t - \theta^t\|_1}_{\text{Term (I)}} + \underbrace{\sum_{t=0}^{T-1} \langle f^t, \theta^t - \overline{\theta}^* \rangle}_{\text{Term (II)}}, \tag{10}$$

where we use Assumption 2.3 such that $\langle f^t, \overline{\theta}^t - \theta^t \rangle \leq \|f^t\|_\infty \|\overline{\theta}^t - \theta^t\|_1 \leq \|\overline{\theta}^t - \theta^t\|_1$. Thus, it suffices to bound the Term (I) and Term (II).

We first show the bound for Term (I). According to Lemma 5.1, by the fact that $L \leq |\mathcal{S}|$ and $|\mathcal{S}|, |\mathcal{A}| \geq 1$, we have that with probability at least $1 - 2\zeta$, the following holds

$$\text{Term (I)} \leq \mathcal{O}\left(L|\mathcal{S}|\sqrt{T|\mathcal{A}|} \log^{\frac{1}{2}}(T|\mathcal{S}||\mathcal{A}|/\zeta)\right). \tag{11}$$

For Term (II), setting $V = L\sqrt{T}, \alpha = LT, \tau = \sqrt{T}$, and $\lambda = 1/T$, by Lemma 5.2, we obtain

$$\text{Term (II)} \leq 8L\sqrt{T|\mathcal{S}||\mathcal{A}|} + \frac{1}{L\sqrt{T}} \sum_{t=0}^{T-1} \langle \mathbf{Q}(t), \mathbf{g}^t(\overline{\theta}^*) - \mathbf{c} \rangle,$$

where we use the inequality that $\log|\mathcal{S}||\mathcal{A}| \le \sqrt{|\mathcal{S}||\mathcal{A}|}$ with the inequality $\sqrt{x} \ge \log x$. Thus, we further need to bound the last term of the above inequality. By Lemma 5.5 and Remark 5.4, with probability at least $1 - 2T\delta$ for all $t \in \{1, \ldots, T\}$, we have

$$\frac{1}{L\sqrt{T}} \sum_{t=0}^{T-1} \langle \mathbf{Q}(t), \mathbf{g}^t(\overline{\theta}^*) - \mathbf{c} \rangle \le \mathcal{O}\big(L|\mathcal{S}|\sqrt{T|\mathcal{A}|} \log^{\frac{3}{2}}(T/\delta)\big),$$

by the facts that $L \le |\mathcal{S}|$, $|\mathcal{S}| > 1$, $|\mathcal{A}| > 1$, and the assumption $T \ge |\mathcal{S}||\mathcal{A}|$, as well as the computation of $\psi$ as $\psi = \mathcal{O}\big(L^2\sqrt{T} + L\log|\mathcal{S}||\mathcal{A}| + L^2\sqrt{T}\log(T|\mathcal{S}||\mathcal{A}|)\big)$. Therefore, with probability at least $1 - 2T\delta$, the following holds

$$\text{Term (II)} \le \mathcal{O}\big(L|\mathcal{S}|\sqrt{T|\mathcal{A}|} \log^{\frac{3}{2}}(T/\delta)\big). \tag{12}$$

Combining (11) and (12) with (10), and letting $\delta = \zeta/T$, by union bound, we eventually obtain that with probability at least $1 - 4\zeta$, the regret bound $\text{Regret}(T) \le \widetilde{\mathcal{O}}\big(L|\mathcal{S}|\sqrt{T|\mathcal{A}|}\big)$ holds, where the notation $\widetilde{\mathcal{O}}(\cdot)$ hides the logarithmic factors. Further let $\zeta \le 1/(4 + 8L/\overline{\sigma}) < 1/4$ (such that $\overline{\sigma}/4 \ge \zeta(\overline{\sigma}/2 + 2L)$ is guaranteed). This completes the proof. $\qquad\square$

## 5.2 Proof of Constraint Violation Bound

**Lemma 5.6.** *The updating rules in Algorithm 1 ensure*

$$\left\| \left[ \sum_{t=0}^{T-1} (\mathbf{g}^t(\theta^t) - \mathbf{c}) \right]_+ \right\|_2 \le \|\mathbf{Q}(T)\|_2 + \sum_{t=1}^{T} \|\theta^t - \theta^{t-1}\|_1.$$

**Lemma 5.7.** *The updating rules in Algorithm 1 ensure*

$$\sum_{t=1}^{T} \|\theta^t - \theta^{t-1}\|_1 \le 2L\sqrt{T|\mathcal{S}||\mathcal{A}|} \log \frac{8T}{|\mathcal{S}||\mathcal{A}|} + \frac{2L}{(1-\lambda)\alpha} \sum_{t=0}^{T-1} \|\mathbf{Q}(t)\|_2 + \frac{2V + \alpha\lambda}{(1-\lambda)\alpha} LT.$$

**Remark 5.8.** *The proof of Lemma 5.7 uses the fact that the confidence interval of $P$ changes only $\sqrt{T|\mathcal{S}||\mathcal{A}|} \log(8T/(|\mathcal{S}||\mathcal{A}|))$ times, thanks to the doubling of epoch length in Algorithm 1. Within each epoch where the confidence interval is unchanged, we further show $\|\theta^t - \theta^{t-1}\|_1$ is small.*

*Proof of Constraint Violation Bound in Theorem 4.3.* We decompose the constraint violation as

$$\left\| \left[ \sum_{t=0}^{T-1} \big(\mathbf{g}^t(\overline{\theta}^t) - \mathbf{c}\big) \right]_+ \right\|_2 \le \sum_{t=0}^{T-1} \|\mathbf{g}^t(\theta^t) - \mathbf{g}^t(\overline{\theta}^t)\|_2 + \left\| \left[ \sum_{t=0}^{T-1} \big(\mathbf{g}^t(\theta^t) - \mathbf{c}\big) \right]_+ \right\|_2$$

$$\le \underbrace{\sum_{t=0}^{T-1} \|\overline{\theta}^t - \theta^t\|_1}_{\text{Term (III)}} + \underbrace{\left\| \left[ \sum_{t=0}^{T-1} \big(\mathbf{g}^t(\theta^t) - \mathbf{c}\big) \right]_+ \right\|_2}_{\text{Term (IV)}}, \tag{13}$$

where the second inequality is due to Assumption 2.3 that $\|\mathbf{g}^t(\theta^t) - \mathbf{g}^t(\overline{\theta}^t)\|_2 = (\sum_{i=1}^{I} |\langle g_i^t, \theta^t - \overline{\theta}^t \rangle|^2)^{\frac{1}{2}} \le \sum_{i=1}^{I} \|g_i^t\|_\infty \|\theta^t - \overline{\theta}^t\|_1 \le \|\theta^t - \overline{\theta}^t\|_1$. Thus, it suffices to bound Terms (III) and (IV).

For Term(III), we already have its bound as (11). Then, we focus on proving the upper bound of Term(IV). Set $V = L\sqrt{T}$, $\alpha = LT$, $\tau = \sqrt{T}$, and $\lambda = 1/T$ as in the proof of the regret bound. By Lemma 5.6, we know that to bound Term(IV) requires bounding the terms $\|\mathbf{Q}(T)\|_2$ and $\sum_{t=1}^{T} \|\theta^t - \theta^{t-1}\|_1$. By Lemma 5.3, combining it with Remark 5.4 and $\psi = \mathcal{O}(L^2\sqrt{T} + L\log|\mathcal{S}||\mathcal{A}| + L^2\log(T|\mathcal{S}||\mathcal{A}|)/\sqrt{T})$ as shown in the proof of the regret bound, letting $\overline{\sigma}/4 \ge \zeta(\overline{\sigma}/2 + 2L)$, with probability $1 - T\delta$, for all $t \in \{1, \ldots, T\}$, the following inequality holds

$$\|\mathbf{Q}(t)\|_2 \le \mathcal{O}\big(L^2\sqrt{T}\log(L/\delta)\big), \tag{14}$$

where we use $\log x \le \sqrt{x}$. This gives the bound of $\|\mathbf{Q}(T)\|_2$ that $\|\mathbf{Q}(T)\|_2 \le \mathcal{O}\big(L^2\sqrt{T}\log(L/\delta)\big)$.

Furthermore, by Lemma 5.7, we know that the the key to bound $\sum_{t=1}^{T}\|\theta^t - \theta^{t-1}\|_1$ is also the drift bound for $\mathbf{Q}(t)$. Therefore, by (14) and the settings of the parameters $\alpha, \lambda, V$, we have

$$\sum_{t=1}^{T}\|\theta^t - \theta^{t-1}\|_1 \leq \mathcal{O}\big(L|\mathcal{S}|\sqrt{|\mathcal{A}|T}\log(T|\mathcal{S}||\mathcal{A}|/\delta)\big), \tag{15}$$

by the facts that $L \leq |\mathcal{S}|$, $|\mathcal{S}| > 1$, $|\mathcal{A}| > 1$ and the condition $|\mathcal{S}||\mathcal{A}| \leq T$. Thus combining (14) and (15) with Lemma 5.6, and letting $\delta = \zeta/T$, then with probability at least $1 - \zeta$, we have

$$\text{Term(IV)} \leq \mathcal{O}\big(L|\mathcal{S}|\sqrt{|\mathcal{A}|T}\log(T|\mathcal{S}||\mathcal{A}|/\delta)\big).$$

Combining results for Term(III) and Term(IV) with (13), by union bound, with probability at least $1 - 4\zeta$, the constraint violation $\text{Violation}(T) \leq \widetilde{\mathcal{O}}\big(L|\mathcal{S}|\sqrt{T|\mathcal{A}|}\big)$ holds. This finishes the proof. $\square$

## 6 Conclusion

In this paper, we propose a new upper confidence primal-dual algorithm to solve online constrained episodic MDPs with arbitrarily varying losses and stochastically changing constraints. In particular, our algorithm does not require transition models of the MDPs and delivers an $\widetilde{\mathcal{O}}(L|\mathcal{S}|\sqrt{|\mathcal{A}|T})$ upper bounds of both the regret and the constraint violation. The analysis builds upon a Lagrange multiplier condition on a sequence of time varying constrained problems. Such a condition enables a new drift analysis making use of the upper confidence bound together with the Lazy update nature of the sequence of confidence interval constructions on the models.

## Broader Impact

The wide application of reinforcement learning urges researchers to design models with certain constraints enforcing the fairness and safety so that the learned policy is under control. In this line of research, there have been many empirical studies focusing on constrained Markov decision process, including autonomous vehicle control, power systems, robotics, and social fairness. However, their theoretical understandings are rather limited. In this paper, we aim at providing theoretical analysis of constrained Markov decision process, helping researchers to better understand how to design effective constrained reinforcement learning algorithms and what their theoretical guarantees are.

## Acknowledgments and Disclosure of Funding

This work was supported by the US National Institute of Health grant 1RF1AG051710-01.

## Footnotes

[1]We let $\operatorname{dist}(\eta, \mathcal{V}_{t,\tau}^*) := \min_{\eta' \in \mathcal{V}_{t,\tau}^*} \frac{1}{2}\|\eta - \eta'\|_2^2$ as Euclidean distance between a point $\eta$ and the set $\mathcal{V}_{t,\tau}^*$.

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
