[Supplementary Material]

# Supplementary Material

## A  Efficient Updating Rules for Subproblem

In this section, we provide the details on how to efficiently solve the subproblem (6). We can further rewrite (6) into the following equivalent form

$$\theta^t = \underset{\theta \in \Delta(\ell(t), \zeta)}{\operatorname{argmin}} \; \alpha^{-1} \langle \psi^{t-1}, \theta \rangle + D(\theta, \widetilde{\theta}^{t-1}),$$

where we let $\psi^{t-1} := Vf^{t-1} + \sum_{i=1}^{I} Q_i(t-1)g_i^{t-1}$. According to Rosenberg and Mansour [2019a], solving the above problem is decomposed to the following two steps

$$\underline{\theta}^t = \underset{\theta}{\operatorname{argmin}} \; \alpha^{-1} \langle \psi^{t-1}, \theta \rangle + D(\theta, \widetilde{\theta}^{t-1}), \tag{16}$$

$$\theta^t = \underset{\theta \in \Delta(\ell(t), \zeta)}{\operatorname{argmin}} \; D(\theta, \underline{\theta}^t). \tag{17}$$

Note that the first step, i.e., (16), is an unconstrained problem, which has a closed-form solution

$$\underline{\theta}^t(s, a, s') = \widetilde{\theta}^{t-1}(s, a, s')e^{-\psi^{t-1}/\alpha}, \quad \forall (s, a, s') \in \mathcal{S}_k \times \mathcal{A} \times \mathcal{S}_{k+1}, \; \forall k = 0, \dots, L-1. \tag{18}$$

The second step, i.e., (17), can be viewed as a projection of $\underline{\theta}^t(s, a, s')$ onto the feasible set $\Delta(\ell(t), \zeta)$. With the definition of the feasible set as in (8), further by Theorem 4.2 of Rosenberg and Mansour [2019a] and Lemma 7 of Jin et al. [2019], and plugging in $\underline{\theta}^t$ computed as (18), we have

$$\theta^t(s, a, s') = \frac{\widetilde{\theta}^{t-1}(s, a, s')}{Z_t^{k(s)}(\mu^t, \beta^t)} e^{B_t^{\mu^t, \beta^t}(s, a, s')}, \tag{19}$$

where $k(s)$ is a mapping for state $s$ to its associated layer index, and $Z_t^k(\mu, \beta)$ and $B_t^{\mu, \beta}$ are defined as follows

$$B_t^{\mu, \beta}(s, a, s') = \mu^-(s, a, s') - \mu^+(s, a, s') + (\mu^+(s, a, s') + \mu^-(s, a, s'))\varepsilon_{\ell(t)}^{\zeta}(s, a) + \beta(s')$$
$$- \beta(s) - \psi^{t-1}(s, a, s')/\alpha - \sum_{s'' \in \mathcal{S}_{k(s)+1}} \widehat{P}_{\ell(t)}(s''|s, a)(\mu^-(s, a, s'') - \mu^+(s, a, s'')),$$

$$Z_t^k(\mu, \beta) = \sum_{s \in \mathcal{S}_k} \sum_{a \in \mathcal{A}} \sum_{s' \in \mathcal{S}_{k+1}} \widetilde{\theta}^{t-1}(s, a, s')e^{B_t^{\mu, \beta}(s, a, s')},$$

where $\beta : \mathcal{S} \to \mathbb{R}$ and $\mu = (\mu^+, \mu^-)$ with $\mu^+, \mu^- : \mathcal{S} \times \mathcal{A} \times \mathcal{S} \to \mathbb{R}_{\geq 0}$. Specifically, the variables $\mu^t, \beta^t$ in (19) are obtained by solving a convex optimization with only non-negativity constraints, which is

$$\mu^t, \beta^t = \underset{\mu, \beta \geq 0}{\operatorname{argmin}} \sum_{k=0}^{L-1} \log Z_t^k(\mu, \beta). \tag{20}$$

Therefore, by solving (20), we can eventually compute $\theta^t$ by (19). Since (20) is associated with a convex optimization with only non-negativity constraints, it can be solved much efficiently.

## B  Structure of Optimization Problem Sequence

We have the following simple sufficient condition which is a direct corollary of Lemma 1 in Nedić and Ozdaglar [2009]:

**Lemma B.1.** *Suppose that the problem* (9) *is feasible. Then, the set of Lagrange multipliers $\mathcal{V}_{t,\tau}^*$ defined in Assumption 4.1 is nonempty and bounded if the Slater condition holds, i.e., $\exists \theta \in \Delta$, $\varepsilon > 0$ such that $\langle g_i, \theta \rangle \leq c_i - \varepsilon$, $\forall i \in \{1, \dots, I\}$.*

In fact, it can be shown that some certain constraint qualification condition more general than Slater condition can imply the boundedness of Lagrange multipliers (see, for example, Lemma 18 of Wei et al. [2019]). According to Wei et al. [2019], Assumption 4.1 is weaker than Slater condition commonly adopted in previous constrained online learning works. The motivation for such a Lagrange multiplier condition is that it is a sufficient condition of a key structural property on the dual function $q^{(t,\tau)}(\eta)$, namely, the error bound condition. Formally, we have the following definition.

**Definition B.2** (Error Bound Condition (EBC)). *Let $h(\mathbf{x})$ be a concave function over $\mathbf{x} \in \mathcal{X}$, where $\mathcal{X}$ is closed and convex. Suppose $\Lambda^* := \operatorname{argmax}_{\mathbf{x} \in \mathcal{X}} h(\mathbf{x})$ is non-empty. The function $h(\mathbf{x})$ satisfies the EBC if there exists constants $\vartheta, \ \sigma > 0$ such that for any $\mathbf{x} \in \mathcal{X}$ satisfying[2] $\operatorname{dist}(\mathbf{x}, \Lambda^*) \geq \vartheta$,*

$$h(\mathbf{x}^*) - h(\mathbf{x}) \geq \sigma \cdot \operatorname{dist}(\mathbf{x}, \Lambda^*) \quad \text{with } \mathbf{x}^* \in \Lambda^*.$$

Note that in Definition B.2, $\Lambda^*$ is a closed convex set, which follows from the fact that $h(\mathbf{x})$ is a concave function and thus all superlevel sets are closed and convex. The following lemma, whose proof can be found in Lemma 5 of Wei et al. [2019], shows the relation between the Lagrange multiplier condition and the dual function.

**Lemma B.3.** *Fix $T \geq 1$. Suppose Assumption 4.1 holds, then for any $t \in \{0, \ldots, T-1\}$ and $\tau = \sqrt{T}$, the dual function $q^{(t,\tau)}(\eta)$ satisfies the EBC with $\sigma > 0$ and $\vartheta > 0$.*

This lemma is equivalent to Lemma 4.2 in the main text.

## C Proofs of Lemmas in Section 5.1

### C.1 Proof of Lemma 5.1

We first provide Lemmas C.1 and C.2 below. Then, we give the proof of Lemma 5.1 based on these lemmas.

**Lemma C.1** (Lemma 19 in Jaksch et al. [2010]). *For any sequence of numbers $x_1, \ldots, x_n$ with $0 \leq x_k \leq X_{k-1} := \max\left\{1, \sum_{i=1}^{k-1} x_i\right\}$ with $1 \leq k \leq n$, the following inequality holds*

$$\sum_{k=1}^{n} \frac{x_k}{\sqrt{X_{k-1}}} \leq (\sqrt{2}+1)\sqrt{X_n}.$$

**Lemma C.2.** *Let $\widehat{d}_t(s)$ and $d_t(s)$ be the state stationary distributions for $\theta^t$ and $\overline{\theta}^t$ respectively, and $\widehat{P}_{\ell(t)}(s'|a,s)$ and $P(s'|a,s)$ be the corresponding transition distributions. Denote $\pi_t(a|s)$ as the policy at episode $t$. There are $\theta^t(s,a,s') = \widehat{d}_t(s)\pi_t(a|s)\widehat{P}_{\ell(t)}(s'|a,s)$ and $\overline{\theta}^t(s,a,s) = d_t(s)\pi_t(a|s)P(s'|a,s)$. On the other hand, there are also $\widehat{d}_t(s') = \sum_{s \in \mathcal{S}_k} \sum_{a \in \mathcal{A}} \theta^t(s,a,s'), \forall s' \in \mathcal{S}_{k+1}$, and $d_t(s') = \sum_{s \in \mathcal{S}_k} \sum_{a \in \mathcal{A}} \theta^t(s,a,s'), \forall s' \in \mathcal{S}_{k+1}$. Then, we have the following inequality*

$$\|\theta^t - \overline{\theta}^t\|_1 \leq \sum_{k=0}^{L-1} \sum_{j=0}^{k} \sum_{s \in \mathcal{S}_j} \sum_{a \in \mathcal{A}} \mu_t(s,a)\|\widehat{P}_{\ell(t)}(\cdot|s,a) - P(\cdot|s,a)\|_1,$$

*where we let $\mu_t(s,a) = d_t(s)\pi_t(a|s)$.*

*Proof of Lemma C.2.* By the definitions of $\widehat{d}_t$, $d_t$, $\widehat{P}_{\ell(t)}$, $P$, and $\pi_t$ shown in Lemma C.2, we have

$$\begin{aligned}
\|\theta^t - \overline{\theta}^t\|_1 &= \sum_{k=0}^{L-1} \sum_{s \in \mathcal{S}_k} \sum_{a \in \mathcal{A}} \|\theta^t(a,s,\cdot) - \overline{\theta}^t(a,s,\cdot)\|_1 \\
&= \sum_{k=0}^{L-1} \sum_{s \in \mathcal{S}_k} \sum_{a \in \mathcal{A}} \pi_t(a|s)\|\widehat{P}_{\ell(t)}(\cdot|a,s)\widehat{d}_t(s) - P(\cdot|a,s)d_t(s)\|_1 \\
&= \sum_{k=0}^{L-1} \sum_{s \in \mathcal{S}_k} \sum_{a \in \mathcal{A}} \pi_t(a|s)\|\widehat{P}_{\ell(t)}(\cdot|a,s)\widehat{d}_t(s) - P(\cdot|a,s)\widehat{d}_t(s) \\
&\qquad\qquad + P(\cdot|a,s)\widehat{d}_t(s) - P(\cdot|a,s)d_t(s)\|_1.
\end{aligned}$$

Thus, with the above equalities, and by triangle inequality for $\|\cdot\|_1$, we can bound the term $\|\theta^t - \overline{\theta}^t\|_1$ in the following way

$$
\begin{aligned}
\|\theta^t - \overline{\theta}^t\|_1 &\leq \sum_{k=0}^{L-1} \sum_{s \in \mathcal{S}_k} \sum_{a \in \mathcal{A}} \pi_t(a|s)[\|\widehat{P}_{\ell(t)}(\cdot|a,s)\widehat{d}_t(s) - P(\cdot|a,s)\widehat{d}_t(s)\|_1 \\
&\quad + \|P(\cdot|a,s)\widehat{d}_t(s) - P(\cdot|a,s)d_t(s)\|_1] \\
&\leq \sum_{k=0}^{L-1} \sum_{s \in \mathcal{S}_k} \sum_{a \in \mathcal{A}} \pi_t(a|s)\widehat{d}_t(s)\|\widehat{P}_{\ell(t)}(\cdot|a,s) - P(\cdot|a,s)\|_1 \\
&\quad + \sum_{k=0}^{L-1} \sum_{s \in \mathcal{S}_k} \sum_{a \in \mathcal{A}} \pi_t(a|s)\|P(\cdot|a,s)\|_1 \cdot |\widehat{d}_t(s) - d_t(s)|.
\end{aligned}
\tag{21}
$$

Then we need to bound the last two terms of (21) respectively. For the first term of RHS in (21), we have

$$
\begin{aligned}
&\sum_{k=0}^{L-1} \sum_{s \in \mathcal{S}_k} \sum_{a \in \mathcal{A}} \pi_t(a|s)\widehat{d}_t(s)\|\widehat{P}_{\ell(t)}(\cdot|a,s) - P(\cdot|a,s)\|_1 \\
&= \sum_{k=0}^{L-1} \sum_{s \in \mathcal{S}_k} \sum_{a \in \mathcal{A}} \mu_t(s,a)\|\widehat{P}_{\ell(t)}(\cdot|a,s) - P(\cdot|a,s)\|_1,
\end{aligned}
\tag{22}
$$

since $\mu_t(s,a) = \pi_t(a|s)d_t(s)$ denotes the joint distribution probability of $(s,a)$.

Next, we bound the last term of RHS in (21), which is

$$
\sum_{k=0}^{L-1} \sum_{s \in \mathcal{S}_k} \sum_{a \in \mathcal{A}} \pi_t(a|s)\|P(\cdot|a,s)\|_1 \cdot |\widehat{d}_t(s) - d_t(s)| = \sum_{k=0}^{L-1} \sum_{s \in \mathcal{S}_k} \sum_{a \in \mathcal{A}} \pi_t(a|s)|\widehat{d}_t(s) - d_t(s)|,
$$

since $\|P(\cdot|a,s)\|_1 = \sum_{s' \in \mathcal{S}_{k+1}} P(s'|a,s) = 1$. Furthermore, we can bound the last term above as

$$
\begin{aligned}
&\sum_{k=0}^{L-1} \sum_{s \in \mathcal{S}_k} \sum_{a \in \mathcal{A}} \pi_t(a|s)|\widehat{d}_t(s) - d_t(s)| \\
&= \sum_{k=0}^{L-1} \sum_{s \in \mathcal{S}_k} |\widehat{d}_t(s) - d_t(s)| \\
&= \sum_{k=1}^{L-1} \sum_{s \in \mathcal{S}_k} |\widehat{d}_t(s) - d_t(s)| \\
&= \sum_{k=1}^{L-1} \sum_{s \in \mathcal{S}_k} \Big| \sum_{s'' \in \mathcal{S}_{k-1}} \sum_{a \in \mathcal{A}} \theta^t(s'',a,s) - \sum_{s'' \in \mathcal{S}_{k-1}} \sum_{a \in \mathcal{A}} \overline{\theta}^t(s'',a,s) \Big|,
\end{aligned}
$$

where the first equality is due to $\sum_{a \in \mathcal{A}} \pi_t(a|s) = 1$, the second equality is due to $\widehat{d}_t(s_0) = d_t(s_0) = 1$, and the third inequality is by the relations $\widehat{d}_t(s) = \sum_{s'' \in \mathcal{S}_{k-1}} \sum_{a \in \mathcal{A}} \theta^t(s'',a,s)$ and $d_t(s) = \sum_{s'' \in \mathcal{S}_{k-1}} \sum_{a \in \mathcal{A}} \overline{\theta}^t(s'',a',s), \forall s \in \mathcal{S}_k$. Further bounding the last term of the above

equation gives

$$\sum_{k=1}^{L-1} \sum_{s \in \mathcal{S}_k} \left| \sum_{s'' \in \mathcal{S}_{k-1}} \sum_{a \in \mathcal{A}} \theta^t(s'', a, s) - \sum_{s'' \in \mathcal{S}_{k-1}} \sum_{a \in \mathcal{A}} \overline{\theta}^t(s'', a, s) \right|$$

$$\leq \sum_{k=1}^{L-1} \sum_{s \in \mathcal{S}_k} \sum_{s'' \in \mathcal{S}_{k-1}} \sum_{a \in \mathcal{A}} \left| \theta^t(s'', a, s) - \overline{\theta}^t(s'', a, s) \right|$$

$$= \sum_{k=1}^{L-1} \sum_{s'' \in \mathcal{S}_{k-1}} \sum_{a \in \mathcal{A}} \left\| \theta^t(s'', a, \cdot) - \overline{\theta}^t(s'', a, \cdot) \right\|_1$$

$$= \sum_{k=0}^{L-2} \sum_{s \in \mathcal{S}_k} \sum_{a \in \mathcal{A}} \left\| \theta^t(s, a, \cdot) - \overline{\theta}^t(s, a, \cdot) \right\|_1,$$

which eventually implies that the last term on RHS of (21) can be bounded as

$$\sum_{k=0}^{L-1} \sum_{s \in \mathcal{S}_k} \sum_{a \in \mathcal{A}} \pi_t(a|s) \|P(\cdot|a,s)\|_1 \cdot |\widehat{d}_t(s) - d_t(s)| \leq \sum_{k=0}^{L-2} \sum_{s \in \mathcal{S}_k} \sum_{a \in \mathcal{A}} \left\| \theta^t(s,a,\cdot) - \overline{\theta}^t(s,a,\cdot) \right\|_1. \tag{23}$$

Therefore, plugging the bounds (22) and (23) in (21), we have

$$\|\theta^t - \overline{\theta}^t\|_1 = \sum_{k=0}^{L-1} \sum_{s \in \mathcal{S}_k} \sum_{a \in \mathcal{A}} \left\| \theta^t(a, s, \cdot) - \overline{\theta}^t(a, s, \cdot) \right\|_1$$

$$\leq \sum_{k=0}^{L-1} \sum_{s \in \mathcal{S}_k} \sum_{a \in \mathcal{A}} \mu_t(s,a) \left\| \widehat{P}_{\ell(t)}(\cdot|a, s) - P(\cdot|a, s) \right\|_1$$

$$+ \sum_{k=0}^{L-2} \sum_{s \in \mathcal{S}_k} \sum_{a \in \mathcal{A}} \left\| \theta^t(s, a, \cdot) - \overline{\theta}^t(s, a, \cdot) \right\|_1.$$

Recursively applying the above inequality, we obtain

$$\|\theta^t - \overline{\theta}^t\|_1 \leq \sum_{k=0}^{L-1} \sum_{j=0}^{k} \sum_{s \in \mathcal{S}_j} \sum_{a \in \mathcal{A}} \mu_t(s,a) \left\| \widehat{P}_{\ell(t)}(\cdot|s, a) - P(\cdot|s, a) \right\|_1,$$

which completes the proof. □

Now, we are in position to give the proof of Lemma 5.1.

*Proof of Lemma 5.1.* The proof for Lemma 5.1 adopts similar ideas in Neu et al. [2012], Rosenberg and Mansour [2019a].

We already know $\widehat{P}_{\ell(t)}(s'|s,a) = \frac{\theta^t(s,a,s')}{\sum_{s' \in S_{k+1}} \theta^t(s,a,s')}$ and $\mu_t(s,a) = \sum_{s' \in \mathcal{S}_{k+1}} \theta^t(s,a,s'), \forall s \in \mathcal{S}_k, a \in \mathcal{A}, s' \in \mathcal{S}_{k+1}, \forall k \in \{0, \ldots, L-1\}$. By Lemma C.2, one can show that

$$\|\theta^t - \overline{\theta}^t\|_1 \leq \sum_{k=0}^{L-1} \sum_{j=0}^{k} \sum_{s \in \mathcal{S}_j} \sum_{a \in \mathcal{A}} \mu_t(s,a) \left\| \widehat{P}_{\ell(t)}(\cdot|s, a) - P(\cdot|s, a) \right\|_1$$

$$= \sum_{k=0}^{L-1} \sum_{j=0}^{k} \sum_{s \in \mathcal{S}_j} \sum_{a \in \mathcal{A}} \left[ (\mu_t(s,a) - \mathbb{I}\{s_j^t = s, \, a_j^t = a\}) \left\| \widehat{P}_{\ell(t)}(\cdot|s, a) - P(\cdot|s, a) \right\|_1 \right.$$

$$\left. + \mathbb{I}\{s_j^t = s, \, a_j^t = a\} \left\| \widehat{P}_{\ell(t)}(\cdot|s, a) - P(\cdot|s, a) \right\|_1 \right],$$

where we denote $\mathbb{I}\{s_j^t = s, \, a_j^t = a\})$ the indicator random variable that equals 1 with probability $\mu_t(s,a), \forall s \in S_j, a \in \mathcal{A}$ and 0 otherwise. Denote $\xi^t(s,a) = \|\widehat{P}_{\ell(t)}(\cdot|s, a) - P(\cdot|s, a)\|_1$ for

abbreviation. We can see that $\xi^t(s,a) \le \|\widehat{P}_{\ell(t)}(\cdot|s,a)\|_1 + \|P(\cdot|s,a)\|_1 = 2$. Summing both sides of the above inequality over $T$ time slots, we obtain

$$\sum_{t=0}^{T-1} \|\theta^t - \overline{\theta}^t\|_1 \le \sum_{t=0}^{T-1} \sum_{k=0}^{L-1} \sum_{j=0}^{k} \sum_{s \in \mathcal{S}_j} \sum_{a \in \mathcal{A}} (\mu_t(s,a) - \mathbb{I}\{s_j^t = s,\ a_j^t = a\})\xi^t(s,a)$$

$$+ \sum_{t=0}^{T-1} \sum_{k=0}^{L-1} \sum_{j=0}^{k} \sum_{s \in \mathcal{S}_j} \sum_{a \in \mathcal{A}} [\mathbb{I}\{s_j^t = s,\ a_j^t = a\}\xi^t(s,a). \tag{24}$$

Next, we bound the first term on RHS of (24). Let $\mathcal{F}^{t-1}$ be the system history up to $(t-1)$-th episode. Then, by the definition of $\mathbb{I}(\cdot,\cdot)$, we have

$$\mathbb{E}\Big\{ \sum_{s \in \mathcal{S}_j} \sum_{a \in \mathcal{A}} (\mu_t(s,a) - \mathbb{I}\{s_j^t = s,\ a_j^t = a\})\xi^t(s,a) \Big| \mathcal{F}^{t-1} \Big\} = 0,$$

since $\xi^t$ is only associated with system randomness history up to $t-1$ episodes. Thus, the term $\sum_{s \in \mathcal{S}_j} \sum_{a \in \mathcal{A}} (\mu_t(s,a) - \mathbb{I}\{s_j^t = s,\ a_j^t = a\})\xi^t(s,a)$ is a martingale difference sequence with respect to $\mathcal{F}^{t-1}$. Furthermore, by $\xi^t(s,a) \le 2$ and $\sum_{s \in \mathcal{S}_j} \sum_{a \in \mathcal{A}} \mathbb{I}\{s_j^t = s,\ a_j^t = a\}) = 1$, there would be

$$\left| \sum_{s \in \mathcal{S}_j} \sum_{a \in \mathcal{A}} (\mu_t(s,a) - \mathbb{I}\{s_j^t = s,\ a_j^t = a\})\xi^t(s,a) \right|$$

$$\le \left| \sum_{s \in \mathcal{S}_j} \sum_{a \in \mathcal{A}} \mathbb{I}\{s_j^t = s,\ a_j^t = a\} \right| \xi^t(s,a) + \left| \sum_{s \in \mathcal{S}_j} \sum_{a \in \mathcal{A}} \mu_t(s,a) \right| \xi^t(s,a) \le 4.$$

Thus, by Azuma's inequality, we obtain that with probability at least $1 - \zeta/L$,

$$\sum_{t=0}^{T-1} \sum_{s \in \mathcal{S}_j} \sum_{a \in \mathcal{A}} (\mu_t(s,a) - \mathbb{I}\{s_j^t = s,\ a_j^t = a\})\xi^t(s,a) \le 4\sqrt{2T \log \frac{L}{\zeta}}.$$

According to union bound, we further have that with probability at least $1 - \zeta$, the above inequality holds for all $j = 0, ..., L-1$. This implies that with probability at least $1 - \zeta$, the following inequality holds

$$\sum_{t=0}^{T-1} \sum_{k=0}^{L-1} \sum_{j=0}^{k} \sum_{s \in \mathcal{S}_j} \sum_{a \in \mathcal{A}} (\mu_t(s,a) - \mathbb{I}\{s_j^t = s,\ a_j^t = a\})\xi^t(s,a) \le 2L^2\sqrt{2T \log \frac{L}{\zeta}}. \tag{25}$$

On the other hand, we adopt the same argument as the first part of the proof of Lemma 5 in Neu et al. [2012] to show the upper bound of $\sum_{t=0}^{T-1} \sum_{k=0}^{L-1} \sum_{j=0}^{k} \sum_{s \in \mathcal{S}_j} \sum_{a \in \mathcal{A}} \mathbb{I}\{s_j^t = s,\ a_j^t = a\}\xi^t(s,a)$ in (24). Recall that $\ell(t)$ denotes the epoch that the $t$-th episode belongs to. By the definition of the state-action pair counter $N_\ell(s,a)$ and $n_\ell(s,a)$, we have

$$N_{\ell(t)}(s,a) = \sum_{q=0}^{\ell(t)-1} n_q(s,a).$$

According to Lemma C.1, we have

$$\sum_{q=1}^{\ell(t)} \frac{n_q(s,a)}{\max\{1, \sqrt{N_q(s,a)}\}} \le (\sqrt{2}+1)\sqrt{N_{\ell(t)}(s,a)}. \tag{26}$$

Since we can rewrite

$$\sum_{t=0}^{T-1} \sum_{k=0}^{L-1} \sum_{j=0}^{k} \sum_{s \in \mathcal{S}_j} \sum_{a \in \mathcal{A}} \mathbb{I}\{s_j^t = s,\ a_j^t = a\}\xi^t(s,a)$$

$$= \sum_{t=0}^{T-1} \sum_{k=0}^{L-1} \sum_{j=0}^{k} \|\widehat{P}_{\ell(t)}(\cdot|s_j^t, a_j^t) - P(\cdot|s_j^t, a_j^t)\|_1,$$

then by Lemma 3.1, the following holds with probability at least $1 - \zeta$,

$$\sum_{t=0}^{T-1}\sum_{k=0}^{L-1}\sum_{j=0}^{k}\sum_{s\in\mathcal{S}_j}\sum_{a\in\mathcal{A}}\mathbb{I}\{s_j^t = s,\ a_j^t = a\}\xi^t(s,a)$$

$$\leq \sum_{k=0}^{L-1}\sum_{j=0}^{k}\sum_{t=0}^{T-1}\sqrt{\frac{2|\mathcal{S}_{j+1}|\log(T|\mathcal{S}||\mathcal{A}|/\zeta)}{\max\{1, N_{\ell(t)}(s_j^t, a_j^t)\}}}$$

$$\leq \sum_{k=0}^{L-1}\sum_{j=0}^{k}\sum_{q=1}^{\ell(T)}\sum_{s\in\mathcal{S}_j}\sum_{a\in\mathcal{A}}n_q(s,a)\sqrt{\frac{2|\mathcal{S}_{j+1}|\log(T|\mathcal{S}||\mathcal{A}|/\zeta)}{\max\{1, N_q(s,a)\}}}$$

$$\leq \sum_{k=0}^{L-1}\sum_{j=0}^{k}\sum_{s\in\mathcal{S}_j}\sum_{a\in\mathcal{A}}(\sqrt{2}+1)\sqrt{2N_{\ell(T)}(s,a)|\mathcal{S}_{j+1}|\log\frac{T|\mathcal{S}||\mathcal{A}|}{\zeta}},$$

where the first inequality is due to Lemma 3.1, the second inequality is by the definition of the global counter $N_{\ell(t)}(s_j^t, a_j^t)$, and the last inequailty is by (26). Thus, further bounding the last term of the above inequality yields

$$\sum_{k=0}^{L-1}\sum_{j=0}^{k}\sum_{s\in\mathcal{S}_j}\sum_{a\in\mathcal{A}}(\sqrt{2}+1)\sqrt{2N_{\ell(T)}(s,a)|\mathcal{S}_{j+1}|\log\frac{T|\mathcal{S}||\mathcal{A}|}{\zeta}}$$

$$\leq \sum_{k=0}^{L-1}\sum_{j=0}^{k}(\sqrt{2}+1)\sqrt{2\sum_{s\in\mathcal{S}_j}\sum_{a\in\mathcal{A}}N_{\ell(T)}(s,a)|\mathcal{S}_j||\mathcal{S}_{j+1}||\mathcal{A}|\log\frac{T|\mathcal{S}||\mathcal{A}|}{\zeta}}$$

$$\leq \sum_{k=0}^{L-1}\sum_{j=0}^{k}(\sqrt{2}+1)\sqrt{2T|\mathcal{S}_j||\mathcal{S}_{j+1}||\mathcal{A}|\log\frac{T|\mathcal{S}||\mathcal{A}|}{\zeta}}$$

$$\leq (\sqrt{2}+1)L|\mathcal{S}|\sqrt{2T|\mathcal{A}|\log\frac{T|\mathcal{S}||\mathcal{A}|}{\zeta}},$$

where the first inequality is due to Jensen's inequality, the second inequality is by the definition of $N_{\ell(T)}(s,a)$ such that $\sum_{s\in\mathcal{S}_j}\sum_{a\in\mathcal{A}}N_{\ell(T)}(s,a) \leq T$, and the last inequality is by bounding the term $\sum_{k=0}^{L-1}\sum_{j=0}^{k}\sqrt{|\mathcal{S}_j||\mathcal{S}_{j+1}|} \leq \sum_{k=0}^{L-1}\sum_{j=0}^{k}(|\mathcal{S}_j| + |\mathcal{S}_{j+1}|)/2 \leq L|\mathcal{S}|$. The above results imply that with probability at least $1 - \zeta$, the following holds

$$\sum_{t=0}^{T-1}\sum_{k=0}^{L-1}\sum_{j=0}^{k}\sum_{s\in\mathcal{S}_j}\sum_{a\in\mathcal{A}}\mathbb{I}\{s_j^t = s,\ a_j^t = a\}\xi^t(s,a) \leq (\sqrt{2}+1)L|\mathcal{S}|\sqrt{2T|\mathcal{A}|\log\frac{T|\mathcal{S}||\mathcal{A}|}{\zeta}}. \quad (27)$$

By union bound, combining (24), (25) and (27), we obtain with probability at least $1 - 2\zeta$,

$$\sum_{t=0}^{T-1}\|\theta^t - \overline{\theta}^t\|_1 \leq (\sqrt{2}+1)L|\mathcal{S}|\sqrt{2T|\mathcal{A}|\log\frac{T|\mathcal{S}||\mathcal{A}|}{\zeta}} + 2L^2\sqrt{2T\log\frac{L}{\zeta}}.$$

This completes the proof. □

## C.2  Proof of Lemma 5.2

We provide Lemmas C.3, C.4, and C.5 first. Then, we give the proof of Lemma 5.2 based on these lemmas.

**Lemma C.3** (Lemma 14 in Wei et al. [2019]). *Let $M$ and $M^o$ denote the probability simplex and the set of the probability simplex excluding the boundary respectively. Assuming $\mathbf{y} \in M^o$, and letting $\mathcal{C} \subseteq M$, then the following inequality holds*

$$h(\mathbf{x}^{opt}) + \alpha D(\mathbf{x}^{opt}, \mathbf{y}) \leq h(\mathbf{z}) + \alpha D(\mathbf{z}, \mathbf{y}) - \alpha D(\mathbf{z}, \mathbf{x}^{opt}),\ \ \forall \mathbf{z} \in \mathcal{C},$$

*where $\mathbf{x}^{opt} \in \arg\min_{\mathbf{x}\in\mathcal{C}} h(\mathbf{x}) + \alpha D(\mathbf{x}, \mathbf{y})$, $h(\cdot)$ is a convex function, and $D(\cdot, \cdot)$ is the unnormalized KL divergence in this paper.*

**Lemma C.4.** *For any $\theta$ and $\theta'$ satisfying $\sum_{s\in\mathcal{S}_k}\sum_{a\in\mathcal{A}}\sum_{s'\in\mathcal{S}_{k+1}}\theta(s,a,s')=1$, and $\theta(s,a,s')\geq 0, \forall k\in\{0,\ldots,L-1\}$ and $\sum_{s\in\mathcal{S}_k}\sum_{a\in\mathcal{A}}\theta(s,a,s')=\sum_{a\in\mathcal{A}}\sum_{s''\in\mathcal{S}_{k+2}}\theta(s',a,s''),\forall s'\in \mathcal{S}_{k+1},\forall k\in\{0,\ldots,L-2\}$, we let $\theta_k := [\theta(s,a,s')]_{s\in\mathcal{S}_k,a\in\mathcal{A},s'\in\mathcal{S}_{k+1}}$ denote the vector formed by the elements $\theta(s,a,s')$ for all $s_k\in\mathcal{S}_k, a_k\in\mathcal{A}, s_{k+1}\in\mathcal{S}_{k+1}$. We also let $\theta'_k := [\theta'(s,a,s')]_{s\in\mathcal{S}_k,a\in\mathcal{A},s'\in\mathcal{S}_{k+1}}$ similarly denote a vector formed by $\theta'(s,a,s')$. Then, we have*

$$D(\theta,\theta')\geq \frac{1}{2}\sum_{k=0}^{L-1}\|\theta_k-\theta'_k\|_1^2\geq \frac{1}{2L}\|\theta-\theta'\|_1^2,$$

*where $D(\cdot,\cdot)$ denotes the un-normalized Bregman divergence.*

*Proof of Lemma C.4.* We prove the lemma by the following inequality

$$D(\theta,\theta')=\sum_{k=0}^{L-1}\sum_{s\in\mathcal{S}_k}\sum_{a\in\mathcal{A}}\sum_{s\in\mathcal{S}_{k+1}}\theta(s,a,s')\frac{\theta(s,a,s')}{\theta'(s,a,s')}-\theta(s,a,s')+\theta'(s,a,s')$$

$$=\sum_{k=0}^{L-1}\sum_{s\in\mathcal{S}_k}\sum_{a\in\mathcal{A}}\sum_{s\in\mathcal{S}_{k+1}}\theta(s,a,s')\frac{\theta(s,a,s')}{\theta'(s,a,s')}$$

$$\geq \frac{1}{2}\sum_{k=0}^{L-1}\|\theta_k-\theta'_k\|_1^2\geq \frac{1}{2L}\left(\sum_{k=0}^{L-1}\|\theta_k-\theta'_k\|_1\right)^2\geq \frac{1}{2L}\|\theta-\theta'\|_1^2,$$

where the inequality is due to the Pinsker's inequality since $\theta_k$ and $\theta'_k$ are two probability distributions such that $\|\theta_k\|_1=1$ and $\|\theta'_k\|_1=1$. This completes the proof. $\qquad\square$

**Lemma C.5.** *For any $\theta$ and $\theta'$ satisfying $\sum_{s\in\mathcal{S}_k}\sum_{a\in\mathcal{A}}\sum_{s'\in\mathcal{S}_{k+1}}\theta(s,a,s')=1$, and $\theta(s,a,s')\geq 0, \forall k\in\{0,\ldots,L-1\}$ and $\sum_{s\in\mathcal{S}_k}\sum_{a\in\mathcal{A}}\theta(s,a,s')=\sum_{a\in\mathcal{A}}\sum_{s''\in\mathcal{S}_{k+2}}\theta(s',a,s''),\forall s'\in \mathcal{S}_{k+1},\forall k\in\{0,\ldots,L-2\}$, letting $\widetilde{\theta}'(s,a,s')=(1-\lambda)\theta'(s,a,s')+\frac{\lambda}{|\mathcal{A}||\mathcal{S}_k||\mathcal{S}_{k+1}|},\forall (s,a,s')\in \mathcal{S}_k\times\mathcal{A}\times\mathcal{S}_{k+1},\forall k=1,\ldots,L-1$ with $0<\lambda\leq 1$, then we have*

$$D(\theta,\widetilde{\theta}')-D(\theta,\theta')\leq \lambda L\log|\mathcal{S}|^2|\mathcal{A}|,$$

$$D(\theta,\widetilde{\theta}')\leq L\log\left(\frac{|\mathcal{S}|^2|\mathcal{A}|}{\lambda}\right).$$

*Proof of Lemma C.5.* We start our proof as follows

$$D(\theta,\widetilde{\theta}')-D(\theta,\theta')=\sum_{k=0}^{L-1}\sum_{s\in\mathcal{S}_k}\sum_{a\in\mathcal{A}}\sum_{s\in\mathcal{S}_{k+1}}\theta(s,a,s')\left(\log\frac{\theta(s,a,s')}{\widetilde{\theta}'(s,a,s')}-\log\frac{\theta(s,a,s')}{\theta'(s,a,s')}\right)$$

$$+\widetilde{\theta}'(s,a,s')-\theta'(s,a,s')$$

$$=\sum_{k=0}^{L-1}\sum_{s\in\mathcal{S}_k}\sum_{a\in\mathcal{A}}\sum_{s\in\mathcal{S}_{k+1}}\theta(s,a,s')\left(\log\theta'(s,a,s')-\log\widetilde{\theta}'(s,a,s')\right)$$

$$=\sum_{k=0}^{L-1}\sum_{s\in\mathcal{S}_k}\sum_{a\in\mathcal{A}}\sum_{s\in\mathcal{S}_{k+1}}\theta(s,a,s')\Big(\log\theta'(s,a,s')$$

$$-\log[(1-\lambda)\theta'(s,a,s')+\lambda/|\mathcal{A}||\mathcal{S}_k||\mathcal{S}_{k+1}|]\Big),$$

where the last equality is by substituting $\widetilde{\theta}'(s, a, s') = (1-\lambda)\theta'(s, a, s') + \frac{\lambda}{|\mathcal{A}||\mathcal{S}_k||\mathcal{S}_{k+1}|}, \forall (s, a, s') \in \mathcal{S}_k \times \mathcal{A} \times \mathcal{S}_{k+1}, \forall k = 1, \ldots, L-1$. Thus, by bounding the last term above, we further have

$$
\begin{aligned}
D(\theta, \widetilde{\theta}') - D(\theta, \theta') \leq & \sum_{k=0}^{L-1} \sum_{s \in \mathcal{S}_k} \sum_{a \in \mathcal{A}} \sum_{s \in \mathcal{S}_{k+1}} \theta(s, a, s') \bigg( \log \theta'(s, a, s') \\
& - (1-\lambda) \log \theta'(s, a, s') - \lambda \log \frac{1}{|\mathcal{S}_k||\mathcal{S}_{k+1}||\mathcal{A}|} \bigg) \\
= & \sum_{k=0}^{L-1} \sum_{s \in \mathcal{S}_k} \sum_{a \in \mathcal{A}} \sum_{s \in \mathcal{S}_{k+1}} \lambda\theta(s, a, s') \big( \log \theta'(s, a, s') + \log(|\mathcal{S}_k||\mathcal{S}_{k+1}||\mathcal{A}|) \big) \\
\leq & \sum_{k=0}^{L-1} \sum_{s \in \mathcal{S}_k} \sum_{a \in \mathcal{A}} \sum_{s \in \mathcal{S}_{k+1}} \lambda\theta(s, a, s') \log(|\mathcal{S}_k||\mathcal{S}_{k+1}||\mathcal{A}|) \leq \lambda L \log |\mathcal{S}|^2 |\mathcal{A}|,
\end{aligned}
$$

where the first inequality is by Jensen's inequality and the second inequality is due to $\log \theta'(s, a, s') \leq 0$ since $0 < \theta(s, a, s') \leq 1$, and the last inequality is due to Hölder's inequality that $\langle \mathbf{x}, \mathbf{y} \rangle \leq \|\mathbf{x}\|_1 \|\mathbf{y}\|_\infty$ and $|\mathcal{S}_k||\mathcal{S}_{k+1}| \leq |\mathcal{S}|^2$.

Moreover, we have

$$
\begin{aligned}
D(\theta, \widetilde{\theta}') = & \sum_{k=0}^{L-1} \sum_{s \in \mathcal{S}_k} \sum_{a \in \mathcal{A}} \sum_{s \in \mathcal{S}_{k+1}} \theta(s, a, s') \log \frac{\theta(s, a, s')}{\widetilde{\theta}'(s, a, s')} - \theta(s, a, s') + \theta'(s, a, s') \\
= & \sum_{k=0}^{L-1} \sum_{s \in \mathcal{S}_k} \sum_{a \in \mathcal{A}} \sum_{s \in \mathcal{S}_{k+1}} \theta(s, a, s') \big( \log \theta(s, a, s') - \log \widetilde{\theta}'(s, a, s') \big) \\
= & \sum_{k=0}^{L-1} \sum_{s \in \mathcal{S}_k} \sum_{a \in \mathcal{A}} \sum_{s \in \mathcal{S}_{k+1}} \theta(s, a, s') \big( \log \theta(s, a, s') - \log[(1-\lambda)\theta'(s, a, s') + \lambda/(|\mathcal{S}_k||\mathcal{S}_{k+1}||\mathcal{A}|)] \big) \\
\leq & - \sum_{k=0}^{L-1} \sum_{s \in \mathcal{S}_k} \sum_{a \in \mathcal{A}} \sum_{s \in \mathcal{S}_{k+1}} \theta(s, a, s') \big( \log[(1-\lambda)\theta'(s, a, s') + \lambda/(|\mathcal{S}_k||\mathcal{S}_{k+1}||\mathcal{A}|)] \big) \\
\leq & - \sum_{k=0}^{L-1} \sum_{s \in \mathcal{S}_k} \sum_{a \in \mathcal{A}} \sum_{s \in \mathcal{S}_{k+1}} \theta(s, a, s') \cdot \log \frac{\lambda}{|\mathcal{S}_k||\mathcal{S}_{k+1}||\mathcal{A}|} \leq L \log \frac{|\mathcal{S}|^2 |\mathcal{A}|}{\lambda},
\end{aligned}
$$

where the first inequality is due to $\log \theta(s, a, s') \leq 0$, the second inequality is due to the monotonicity of logarithm function, and the third inequality is by as well as $|\mathcal{S}_k||\mathcal{S}_{k+1}| \leq |\mathcal{S}|^2$. This completes the proof. □

Now we are ready to provide the proof of Lemma 5.2.

*Proof of Lemma 5.2.* First of all, by Lemma 3.1, we know that

$$
\|P(\cdot|s, a) - \widehat{P}_\ell(\cdot|s, a)\|_1 \leq \varepsilon_\ell^\zeta(s, a),
$$

with probability at least $1 - \zeta$, for all epochs $\ell$ and any state and action pair $(s, a) \in \mathcal{S} \times \mathcal{A}$. Thus, we have that for any epoch $\ell \leq \ell(T)$,

$$
\Delta \subseteq \Delta(\ell, \zeta)
$$

holds with probability at least $1 - \zeta$.

This can be easily proved in the following way: If any $\bar{\theta} \in \Delta$, then for all $k = \{0, \ldots, T-1\}$, $s \in \mathcal{S}_k$ and $a \in \mathcal{A}$,

$$
\frac{\bar{\theta}(s, a, \cdot)}{\sum_{s' \in \mathcal{S}_{k+1}} \bar{\theta}(s, a, s')} = P(\cdot|s, a).
$$

Then, we obtain with probability at least $1 - \zeta$,

$$
\left\| \frac{\overline{\theta}(s,a,\cdot)}{\sum_{s' \in \mathcal{S}_{k+1}} \overline{\theta}(s,a,s')} - \widehat{P}_\ell(\cdot|s,a) \right\|_1
$$

$$
\leq \left\| \frac{\overline{\theta}(s,a,\cdot)}{\sum_{s' \in \mathcal{S}_{k+1}} \overline{\theta}(s,a,s')} - P(\cdot|s,a) \right\|_1 + \left\| P(\cdot|s,a) - \widehat{P}_\ell(\cdot|s,a) \right\|_1
$$

$$
\leq 0 + \varepsilon_\ell^\zeta(s,a) \leq \varepsilon_\ell^\zeta(s,a).
$$

where the last inequality is by Lemma 3.1. Therefore, we know that $\overline{\theta} \in \Delta(\ell, \zeta)$, which proves the above claim.

Therefore, we define the event as follows

$$
\text{Event } \mathcal{D}_T : \Delta \subseteq \cap_{\ell=1}^{\ell(T)} \Delta(\ell, \zeta), \tag{28}
$$

by which we have

$$
Pr(\mathcal{D}_T) \geq 1 - \zeta.
$$

Thus, for any $\overline{\theta}^*$ that is a solution to problem (1), we have $\overline{\theta}^* \in \Delta$. If event $\mathcal{D}_T$ happens, then $\overline{\theta}^* \in \cap_{\ell=1}^{\ell(T)} \Delta(\ell, \zeta)$. Now we have that the updating rule of $\theta$ follows $\theta^t = \arg\min_{\theta \in \Delta(\ell(t), \zeta)} \langle V f^{t-1} + \sum_{i=1}^{I} Q_i(t-1) g_i^{t-1}, \theta \rangle + \alpha D(\theta, \widetilde{\theta}^{t-1})$ as shown in (6), and also $\overline{\theta}^* \in \cap_{\ell=1}^{\ell(T)} \Delta(\ell, \zeta), \forall \ell$ holds with probability at least $1 - \zeta$. According to Lemma C.3, letting $\mathbf{x}^{opt} = \theta^t$, $\mathbf{z} = \overline{\theta}^*$, $\mathbf{y} = \widetilde{\theta}^{t-1}$ and $h(\theta) = \langle V f^{t-1} + \sum_{i=1}^{I} Q_i(t-1) g_i^{t-1}, \theta \rangle$, we have that with probability at least $1 - \zeta$, the following holds for all epochs $t = 1, \ldots, T$

$$
\left\langle V f^{t-1} + \sum_{i=1}^{I} Q_i(t-1) g_i^{t-1}, \theta^t \right\rangle + \alpha D(\theta^t, \widetilde{\theta}^{t-1})
$$

$$
\leq \left\langle V f^{t-1} + \sum_{i=1}^{I} Q_i(t-1) g_i^{t-1}, \overline{\theta}^* \right\rangle + \alpha D(\overline{\theta}^*, \widetilde{\theta}^{t-1}) - \alpha D(\overline{\theta}^*, \theta^t), \tag{29}
$$

which means once given the event $\mathcal{D}_T$ happens, the inequality (29) will hold.

On the other hand, according to the updating rule of $\mathbf{Q}(\cdot)$ in (5), which is $Q_i(t) = \max\{Q_i(t-1) + \langle g_i^{t-1}, \theta^t \rangle - c_i, 0\}$, we know that

$$
Q_i(t)^2 \leq \left( \max\{Q_i(t-1) + \langle g_i^{t-1}, \theta^t \rangle - c_i, 0\} \right)^2 \leq \left( Q_i(t-1) + \langle g_i^{t-1}, \theta^t \rangle - c_i \right)^2,
$$

which further leads to

$$
Q_i(t)^2 - Q_i(t-1)^2 \leq 2 Q_i(t-1) \left( \langle g_i^{t-1}, \theta^t \rangle - c_i \right) + \left( \langle g_i^{t-1}, \theta^t \rangle - c_i \right)^2.
$$

Taking summation on both sides of the above inequality from $i = 1$ to $I$, we have

$$
\frac{1}{2} \left( \|\mathbf{Q}(t)\|^2 - \|\mathbf{Q}(t-1)\|^2 \right)
$$

$$
\leq \sum_{i=1}^{I} \langle Q_i(t-1) g_i^{t-1}, \theta^t \rangle - \sum_{i=1}^{I} Q_i(t-1) c_i + \frac{1}{2} \sum_{i=1}^{I} \left( \langle g_i^{t-1}, \theta^t \rangle - c_i \right)^2 \tag{30}
$$

$$
\leq \sum_{i=1}^{I} \langle Q_i(t-1) g_i^{t-1}, \theta^t \rangle - \sum_{i=1}^{I} Q_i(t-1) c_i + 2L^2,
$$

where we let $\|\mathbf{Q}(t)\|^2 = \sum_{i=1}^{I} Q_i^2(t)$ and $\|\mathbf{Q}(t-1)\|^2 = \sum_{i=1}^{I} Q_i^2(t-1)$, and the last inequality is due to

$$\sum_{i=1}^{I}(\langle g_i^{t-1}, \theta^t\rangle - c_i)^2 \leq 2\sum_{i=1}^{I}[(\langle g_i^{t-1}, \theta^t\rangle)^2 + c_i^2]$$

$$\leq 2\sum_{i=1}^{I}[\|g_i^{t-1}\|_\infty^2\|\theta^t\|_1^2 + c_i^2]$$

$$\leq 2\sum_{i=1}^{I}[L^2\|g_i^{t-1}\|_\infty^2 + c_i^2]$$

$$\leq 2[L^2(\sum_{i=1}^{I}\|g_i^{t-1}\|_\infty)^2 + (\sum_{i=1}^{I}|c_i|)^2] \leq 4L^2$$

by Assumption 2.3 and the facts that $\sum_{s\in\mathcal{S}_k}\sum_{a\in\mathcal{A}}\sum_{s'\in\mathcal{S}_{k+1}}\theta^t(s,a,s') = 1$ and $\theta^t(s,a,s') \geq 0$. Thus, summing up (29) and (30), and then subtracting $\langle Vf^{t-1}, \theta^{t-1}\rangle$ from both sides, we have

$$V\langle f^{t-1}, \theta^t - \theta^{t-1}\rangle + \frac{1}{2}\left(\|\mathbf{Q}(t)\|^2 - \|\mathbf{Q}(t-1)\|^2\right) + \alpha D(\theta^t, \widetilde{\theta}^{t-1})$$

$$\leq V\langle f^{t-1}, \overline{\theta}^* - \theta^{t-1}\rangle + \sum_{i=1}^{I} Q_i(t-1)(\langle g_i^{t-1}, \overline{\theta}^*\rangle - c_i) + \alpha D(\overline{\theta}^*, \widetilde{\theta}^{t-1}) - \alpha D(\overline{\theta}^*, \theta^t) + 4L^2.$$

We further need to show the lower bound of the term $V\langle f^{t-1}, \theta^t - \theta^{t-1}\rangle + \alpha D(\theta^t, \widetilde{\theta}^{t-1})$ on LHS of the above inequality. Specifically, we have

$$V\langle f^{t-1}, \theta^t - \theta^{t-1}\rangle + \alpha D(\theta^t, \widetilde{\theta}^{t-1})$$

$$= V\langle f^{t-1}, \theta^t - \widetilde{\theta}^{t-1}\rangle + V\langle f^{t-1}, \widetilde{\theta}^{t-1} - \theta^{t-1}\rangle + \alpha D(\theta^t, \widetilde{\theta}^{t-1})$$

$$\geq -V\|f^{t-1}\|_\infty \cdot \|\theta^t - \widetilde{\theta}^{t-1}\|_1 - V\|f^{t-1}\|_\infty \cdot \|\widetilde{\theta}^{t-1} - \theta^{t-1}\|_1 + \frac{\alpha}{2}\sum_{k=0}^{L-1}\|\theta_k^t - \widetilde{\theta}_k^{t-1}\|_1^2$$

$$\geq -V\sum_{k=0}^{L-1}\|\theta_k^t - \widetilde{\theta}_k^{t-1}\|_1 - 2L\lambda V + \frac{\alpha}{2}\sum_{k=0}^{L-1}\|\theta_k^t - \widetilde{\theta}_k^{t-1}\|_1^2$$

$$\geq -\frac{LV}{2\alpha} - 2L\lambda V,$$

where the first inequality uses Hölder's inequality and Lemma C.4 that $D(\theta, \theta') = \sum_{k=1}^{L} D(\theta_k, \theta_k') \geq \frac{1}{2}\sum_{k=1}^{L}\|\theta_k - \theta_k'\|_1^2$ with $\theta_k := [\theta(s,a,s')]_{s_k\in\mathcal{S}_k, a_k\in\mathcal{A}, s_{k+1}\in\mathcal{S}_{k+1}}$, the second inequality is due to $\widetilde{\theta}_k^{t-1} = (1-\lambda)\theta_k^{t-1} + \lambda\frac{1}{|\mathcal{A}||\mathcal{S}_k||\mathcal{S}_{k+1}|}$, the second inequality is due to $\|\widetilde{\theta}^{t-1} - \theta^{t-1}\|_1 = \sum_{k=0}^{L-1}\|\widetilde{\theta}_k^{t-1} - \theta_k^{t-1}\|_1 = \lambda\sum_{k=0}^{L-1}\|\theta_k^{t-1} - \frac{1}{|\mathcal{A}||\mathcal{S}_k||\mathcal{S}_{k+1}|}\|_1 \leq \lambda\sum_{k=0}^{L-1}(\|\theta_k^{t-1}\|_1 + \|\frac{1}{|\mathcal{A}||\mathcal{S}_k||\mathcal{S}_{k+1}|}\|_1) \leq 2\lambda L$, and the third inequality is by finding the minimal value of a quadratic function $-Vx + \frac{\alpha}{2}x^2$.

Therefore, one can show that with probability at least $1 - \zeta$, the following inequality holds for all epochs $\ell > 0$,

$$\frac{1}{2}\left(\|\mathbf{Q}(t)\|^2 - \|\mathbf{Q}(t-1)\|^2\right) - \frac{LV}{2\alpha} - 2L\lambda V \tag{31}$$

$$\leq V\langle f^{t-1}, \overline{\theta}^* - \theta^{t-1}\rangle + \sum_{i=1}^{I} Q_i(t-1)(\langle g_i^{t-1}, \overline{\theta}^*\rangle - c_i) + \alpha D(\overline{\theta}^*, \widetilde{\theta}^{t-1}) - \alpha D(\overline{\theta}^*, \theta^t) + 4L^2.$$

Note that according to Lemma C.5, we have

$$D(\overline{\theta}^*, \widetilde{\theta}^{t-1}) - D(\overline{\theta}^*, \theta^t) = D(\overline{\theta}^*, \widetilde{\theta}^{t-1}) - D(\overline{\theta}^*, \theta^{t-1}) + D(\overline{\theta}^*, \theta^{t-1}) - D(\overline{\theta}^*, \theta^t)$$

$$\leq \lambda L \log|\mathcal{S}|^2|\mathcal{A}| + D(\overline{\theta}^*, \theta^{t-1}) - D(\overline{\theta}^*, \theta^t).$$

Therefore, plugging the above inequality into (31) and rearranging the terms, we further get

$$V \left\langle f^{t-1}, \theta^{t-1} - \overline{\theta}^* \right\rangle \leq \frac{1}{2} \left( \|\mathbf{Q}(t-1)\|^2 - \|\mathbf{Q}(t)\|^2 \right) + \sum_{i=1}^{I} Q_i(t-1)(\langle g_i^{t-1}, \overline{\theta}^* \rangle - c_i)$$
$$+ \alpha\lambda L \log |\mathcal{S}|^2 |\mathcal{A}| + \alpha D(\overline{\theta}^*, \theta^{t-1}) - \alpha D(\overline{\theta}^*, \theta^t) + 4L^2 + \frac{LV}{2\alpha} + 2L\lambda V.$$

Thus, by taking summation on both sides of the above inequality from 1 to $T$ and assuming $\mathbf{Q}(0) = 0$, we would obtain that with probability at least $1 - \zeta$, the following inequality holds

$$\sum_{t=1}^{T} \left\langle f^{t-1}, \theta^{t-1} - \overline{\theta}^* \right\rangle \leq \frac{1}{V} \sum_{t=1}^{T} \sum_{i=1}^{I} Q_i(t-1)(\langle g_i^{t-1}, \overline{\theta}^* \rangle - c_i) + \frac{T\alpha\lambda L \log |\mathcal{S}|^2 |\mathcal{A}|}{V}$$
$$+ \frac{\alpha D(\overline{\theta}^*, \theta^0) + 4L^2 T}{V} + \frac{LT}{2\alpha} + 2L\lambda T. \tag{32}$$

It is not difficult to compute that $D(\overline{\theta}^*, \theta^0) \leq L \log |\mathcal{S}|^2 |\mathcal{A}|$ according to the initialization of $\theta^0$ by the uniform distribution. Then, by rearranging the terms, we rewrite (32) as

$$\sum_{t=0}^{T-1} \left\langle f^t, \theta^t - \overline{\theta}^* \right\rangle$$
$$\leq \frac{1}{V} \sum_{t=1}^{T} \sum_{i=1}^{I} Q_i(t)(\langle g_i^t, \overline{\theta}^* \rangle - c_i) + \frac{4L^2 T + (\lambda T + 1)\alpha L \log |\mathcal{S}|^2 |\mathcal{A}|}{V} + \frac{LT}{2\alpha} + 2L\lambda T.$$

This completes the proof. □

## C.3    Proof of Lemma 5.3

In thie subsection, we first provide Lemmas C.6 below. Then, we give the proof of Lemma 5.3 based on these lemmas.

**Lemma C.6** (Lemma 5 of Yu et al. [2017])**.** *Let $\{Z(t), t \geq 0\}$ be a discrete time stochastic process adapted to a filtration $\{\mathcal{F}^t, t \geq 0\}$ with $Z(0) = 0$ and $\mathcal{F}^0 = \{\emptyset, \Omega\}$. Suppose there exists an integer $\tau > 0$, real constants $\theta > 0$, $\rho_{\max} > 0$ and $0 < \kappa \leq \rho_{\max}$ such that*

$$|Z(t+1) - Z(t)| \leq \rho_{\max},$$
$$\mathbb{E}[Z(t+\tau) - Z(t)|\mathcal{F}^t] \leq \begin{cases} \tau\rho_{\max}, & \text{if } Z(t) < \psi \\ -\tau\kappa, & \text{if } Z(t) \geq \psi \end{cases}$$

*hold for all $t \in \{1, 2, ...\}$. Then for any constant $0 < \delta < 1$, with probability at least $1 - \delta$, we have*

$$Z(t) \leq \psi + \tau \frac{4\rho_{\max}^2}{\kappa} \log \left( 1 + \frac{8\rho_{\max}^2}{\kappa^2} e^{\kappa/(4\rho_{\max})} \right) + \tau \frac{4\rho_{\max}^2}{\kappa} \log \frac{1}{\delta}, \quad \forall t \in \{1, 2, ...\}.$$

Now, we are in position to give the proof of Lemma 5.3.

*Proof of Lemma 5.3.* The proof of this Lemma is based on applying the lemma C.6 to our problem. Thus, this proof mainly focuses on showing that the variable $\|\mathbf{Q}(t)\|_2$ satisfies the condition of Lemma C.6.

According to the updating rule of $Q_i(t)$, which is $Q_i(t+1) = \max\{Q_i(t) + \langle g_i^t, \theta^{t+1} \rangle - c_i, 0\}$, we have

$$|\|\mathbf{Q}(t+1)\|_2 - \|\mathbf{Q}(t)\|_2| \leq \|\mathbf{Q}(t+1) - \mathbf{Q}(t)\|_2$$
$$= \sqrt{\sum_{i=1}^{I} |Q_i(t+1) - Q_i(t)|^2}$$
$$\leq \sqrt{\sum_{i=1}^{I} |\langle g_i^t, \theta^{t+1} \rangle - c_i|^2},$$

where the first inequality is due to triangle inequality, and the second inequality is by the fact that $|\max\{a+b,0\}-a|\le|b|$ if $a\ge0$. Then, by Assumption 2.3, we further have

$$\sqrt{\sum_{i=1}^{I}|\langle g_i^t,\theta^{t+1}\rangle-c_i|^2}\le\sum_{i=1}^{I}|\langle g_i^t,\theta^{t+1}\rangle-c_i|\le\sum_{i=1}^{I}(\|g_i^t\|_\infty\|\theta^{t+1}\|_1+|c_i|)\le2L,$$

which therefore implies

$$|\|\mathbf{Q}(t+1)\|_2-\|\mathbf{Q}(t)\|_2|\le2L.$$

Thus, with the above inequality, we have

$$\|\mathbf{Q}(t+\tau)\|_2-\|\mathbf{Q}(t)\|_2\le|\|\mathbf{Q}(t+\tau)\|_2-\|\mathbf{Q}(t)\|_2|$$
$$\le\sum_{\tau=1}^{\tau}|\|\mathbf{Q}(t+\tau)\|_2-\|\mathbf{Q}(t+\tau-1)\|_2|\tag{33}$$
$$\le2\tau L,$$

such that

$$\mathbb{E}[\|\mathbf{Q}(t+\tau)\|_2-\|\mathbf{Q}(t)\|_2|\mathcal{F}^t]\le2\tau L.\tag{34}$$

Note that (33) in fact indicates that the random process $\|\mathbf{Q}(t+\tau)\|_2-\|\mathbf{Q}(t)\|_2$ is bounded by the value $2\tau L$.

Next, we need to show that there exist $\psi$ and $\kappa$ such that $\mathbb{E}[\|\mathbf{Q}(t+\tau)\|_2-\|\mathbf{Q}(t)\|_2|\mathcal{F}^t]\le-\tau\kappa$ if $\|\mathbf{Q}(t)\|_2\ge\psi$. Recall the definition of the event $\mathcal{D}_T$ in (28). Therefore, we have that with probability at least $1-\zeta$, the event $\mathcal{D}_T$ happens, such that for all $t'=1,...,T$ and any $\theta\in\cap_{\ell=1}^{\ell(T)}\Delta(\ell,\zeta)$, the following holds

$$V\langle f^{t'-1},\theta^{t'-1}-\overline{\theta}^*\rangle\le\frac{1}{2}\left(\|\mathbf{Q}(t'-1)\|_2^2-\|\mathbf{Q}(t')\|_2^2\right)+\sum_{i=1}^{I}Q_i(t'-1)(\langle g_i^{t'-1},\theta\rangle-c_i)$$
$$+\alpha\lambda L\log|\mathcal{S}|^2|\mathcal{A}|+\alpha D(\theta,\widetilde{\theta}^{t'-1})-\alpha D(\theta,\theta^{t'})+4L^2+\frac{LV}{2\alpha}+2L\lambda V,$$

which adopts similar proof techniques to (31). Then, the above inequality further leads to the following inequality by rearranging the terms

$$\|\mathbf{Q}(t')\|_2^2-\|\mathbf{Q}(t'-1)\|_2^2\le-2V\langle f^{t'-1},\theta^{t'-1}-\theta\rangle+2\sum_{i=1}^{I}Q_i(t'-1)(\langle g_i^{t'-1},\theta\rangle-c_i)$$
$$+2\alpha\lambda L\log|\mathcal{S}|^2|\mathcal{A}|+2\alpha D(\theta,\widetilde{\theta}^{t'-1})-2\alpha D(\theta,\theta^{t'})+8L^2+\frac{LV}{\alpha}+4L\lambda V.$$

Taking summation from $t+1$ to $\tau+t$ on both sides of the above inequality, and by union bound, the following inequality holds with probability $1-\zeta$ for $\tau=\sqrt{T}$ and $t$ satisfying $0\le t+\tau\le T$

$$\|\mathbf{Q}(\tau+t)\|_2^2-\|\mathbf{Q}(t)\|_2^2$$
$$\le-2V\sum_{t'=t+1}^{\tau+t}\langle f^{t'-1},\theta^{t'-1}-\theta\rangle+2\sum_{t'=t+1}^{\tau+t}\sum_{i=1}^{I}Q_i(t'-1)(\langle g_i^{t'-1},\theta\rangle-c_i)+2\alpha D(\theta,\widetilde{\theta}^t)\tag{35}$$
$$-2\alpha D(\theta,\widetilde{\theta}^{\tau+t})+\sum_{t'=t+1}^{\tau+t}2\alpha[D(\theta,\widetilde{\theta}^{t'-1})-D(\theta,\theta^{t'-1})]+8\tau L^2+\frac{\tau LV}{\alpha}+4\tau L\lambda V.$$

Particularly, in (35), the term $-2\alpha D(\theta,\theta^{t'-1})\le0$ due to the non-negativity of Bregman divergence. By Lemma C.5, we can bound

$$\sum_{\tau=t+1}^{\tau+t}2\alpha[D(\theta,\widetilde{\theta}^{t'-1})-D(\theta,\theta^{t'-1})]\le2\alpha\tau L\log|\mathcal{S}|^2|\mathcal{A}|.$$

For the term $2\alpha D(\theta, \widetilde{\theta}^t)$, by Lemma C.5, we can bound it as

$$2\alpha D(\theta, \widetilde{\theta}^t) \le 2\alpha L \log(|\mathcal{S}|^2|\mathcal{A}|/\lambda).$$

Moreover, we can decompose the term $2V \sum_{t'=t+1}^{\tau+t} \langle f^{t'-1}, \theta - \theta^{t'-1}\rangle + 2\sum_{t'=t+1}^{\tau+t}\sum_{i=1}^{I} Q_i(t'-1)(\langle g_i^{t'-1}, \overline{\theta}^*\rangle - c_i)$ in (35) as

$$
2V \sum_{t'=t+1}^{\tau+t} \left\langle f^{t'-1}, \theta - \theta^{t'-1}\right\rangle + 2\sum_{t'=t+1}^{\tau+t}\sum_{i=1}^{I} Q_i(t'-1)(\langle g_i^{t'-1}, \theta\rangle - c_i)
$$

$$
= 2V \sum_{t'=t+1}^{\tau+t} \left\langle f^{t'-1}, \theta - \theta^{t'-1}\right\rangle + 2\sum_{i=1}^{I} Q_i(t) \sum_{t'=t+1}^{\tau+t} (\langle g_i^{t'-1}, \theta\rangle - c_i)
$$

$$
+ 2 \sum_{t'=t+2}^{\tau+t}\sum_{i=1}^{I}[Q_i(t'-1) - Q_i(t)](\langle g_i^{t'-1}, \theta\rangle - c_i)
$$

$$
\le 2V \sum_{t'=t+1}^{\tau+t} \left\langle f^{t'-1}, \theta\right\rangle + 2\sum_{i=1}^{I} Q_i(t) \sum_{t'=t+1}^{\tau+t} (\langle g_i^{t'-1}, \theta\rangle - c_i) + 2L\tau^2 + 2VL\tau,
$$

where the last inequality is due to

$$
-2V \sum_{t'=t+1}^{\tau+t} \left\langle f^{t'-1}, \theta^{t'-1}\right\rangle \le 2V \sum_{t'=t+1}^{\tau+t}\sum_{k=0}^{L-1}\sum_{s\in\mathcal{S}_k}\sum_{a\in\mathcal{A}}\sum_{s'\in\mathcal{S}_{k+1}} f^{t'-1}(s,a,s')\theta^{t'-1}(s,a,s') \le 2VL\tau,
$$

as well as

$$
2 \sum_{t'=t+2}^{\tau+t}\sum_{i=1}^{I}[Q_i(t'-1) - Q_i(t)](\langle g_i^{t'-1}, \theta\rangle - c_i)
$$

$$
\le 2 \sum_{t'=t+2}^{\tau+t}\sum_{i=1}^{I}\sum_{r=t}^{t'-2} |\langle g_i^r, \theta^{r+1}\rangle - c_i| \cdot |\langle g_i^{t'-1}, \theta\rangle - c_i|
$$

$$
\le \sum_{t'=t+2}^{\tau+t}\sum_{r=t}^{t'-2} \sqrt{\sum_{i=1}^{I} |\langle g_i^r, \theta^{r+1}\rangle - c_i|^2} + \sum_{t'=t+2}^{\tau+t}\sum_{r=t}^{t'-2} \sqrt{\sum_{i=1}^{I} |\langle g_i^{t'-1}, \theta\rangle - c_i|^2}
$$

$$
\le 2L\tau^2,
$$

by $Q_i(t+1) = \max\{Q_i(t) + \langle g_i^t, \theta^{t+1}\rangle - c_i, 0\}$ and $|\max\{a+b,0\} - a| \le |b|$ if $a \ge 0$ for the first inequality, and Assumption 2.3 for the last inequality.

Therefore, taking conditional expectation on both sides of (35) and combining the above upper bounds for certain terms in (35), we can obtain

$$
\mathbb{E}[\|\mathbf{Q}(\tau+t)\|^2 - \|\mathbf{Q}(t)\|^2 | \mathcal{F}^t, \mathcal{D}_T]
$$

$$
\le 2\tau^2 L + 2\alpha L \log(|\mathcal{S}|^2|\mathcal{A}|/\lambda)
$$

$$
+ 2V\tau\mathbb{E}\left[\frac{1}{\tau}\sum_{t'=t+1}^{\tau+t}\langle f^{t'-1}, \theta\rangle + \frac{1}{\tau}\sum_{i=1}^{I}\frac{Q_i(t)}{V}\sum_{t'=t+1}^{\tau+t}(\langle g_i^{t'-1}, \theta\rangle - c_i)\bigg| \mathcal{F}^t, \mathcal{D}_T\right] \quad (36)
$$

$$
+ 2\alpha\lambda\tau L \log|\mathcal{S}|^2|\mathcal{A}| + 8\tau L^2 + \frac{\tau LV}{\alpha} + 4\tau L\lambda V + 2VL\tau.
$$

Thus, it remains to bound the term $\mathbb{E}[\frac{1}{\tau}\sum_{t'=t+1}^{\tau+t}\langle f^{t'-1}, \theta\rangle + \frac{1}{\tau}\sum_{i=1}^{I}\frac{Q_i(t)}{V}\sum_{t'=t+1}^{\tau+t}(\langle g_i^{t'-1}, \theta\rangle - c_i)|\mathcal{F}^t, \mathcal{D}_T]$ so as to give an upper bound of the right-hand side of (36). Given the event $\mathcal{D}_T$ happens such that $\Delta \subseteq \cap_{\ell=1}^{\ell(T)}\Delta(\ell, \varsigma) \ne \emptyset$, and since $\theta$ is any vector in the set $\cap_{\ell=1}^{\ell(T)}\Delta(\ell, \varsigma)$, we can give an

upper bound of (36) by bounding a term $q^{(t,\tau)}\left(\frac{\mathbf{Q}(t)}{V}\right)$, which is formulated as

$$\min_{\theta\in\cap_{\ell=1}^{\ell(T)}\Delta(\ell,\zeta)}\mathbb{E}\left[\frac{1}{\tau}\sum_{t'=t+1}^{\tau+t}\left\langle f^{t'-1},\theta\right\rangle+\frac{1}{\tau}\sum_{i=1}^{I}\frac{Q_i(t)}{V}\sum_{t'=t+1}^{\tau+t}\left(\langle g_i^{t'-1},\theta\rangle-c_i\right)\Big|\mathcal{F}^t,\mathcal{D}_T\right]$$

$$=\min_{\theta\in\cap_{\ell=1}^{\ell(T)}\Delta(\ell,\zeta)}\left\langle f^{(t,\tau)},\theta\right\rangle+\sum_{i=1}^{I}\frac{Q_i(t)}{V}(\langle g_i,\theta\rangle-c_i)$$

$$\leq\min_{\theta\in\Delta}\left\langle f^{(t,\tau)},\theta\right\rangle+\sum_{i=1}^{I}\frac{Q_i(t)}{V}(\langle g_i,\theta\rangle-c_i)$$

$$=q^{(t,\tau)}\left(\frac{\mathbf{Q}(t)}{V}\right),$$

where the inequality is due to $\Delta\subseteq\cap_{\ell=1}^{\ell(T)}\Delta(\ell,\zeta)$ given $\mathcal{D}_T$ happens and the last equality is obtained according to the definition of the dual function $q$ in Section 4. We can bound $q^{(t,\tau)}\left(\frac{\mathbf{Q}(t)}{V}\right)$ in the following way.

According to Assumption 4.1, we assume that one dual solution is $\eta_{t,\tau}^*\in\mathcal{V}_{t,\tau}^*$. We let $\overline{\vartheta}$ be the maximum of all $\vartheta$ and $\overline{\sigma}$ be the minimum of all $\sigma$. Thus, when $\mathrm{dist}(\frac{\mathbf{Q}(t)}{V},\mathcal{V}_{t,\tau}^*)\geq\overline{\vartheta}$, we have

$$q^{(t,\tau)}\left(\frac{\mathbf{Q}(t)}{V}\right)=q^{(t,\tau)}\left(\frac{\mathbf{Q}(t)}{V}\right)-q^{(t,\tau)}(\eta_{t,\tau}^*)+q^{(t,\tau)}(\eta_{t,\tau}^*)$$

$$\leq-\overline{\sigma}\left\|\eta_{t,\tau}^*-\frac{\mathbf{Q}(t)}{V}\right\|_2+\left\langle f^{(t,\tau)},\theta_{t,\tau}^*\right\rangle$$

$$\leq-\overline{\sigma}\left\|\frac{\mathbf{Q}(t)}{V}\right\|_2+\overline{\sigma}\|\eta_{t,\tau}^*\|_2+\sum_{k=0}^{L-1}\sum_{s\in\mathcal{S}_k}\sum_{a\in\mathcal{A}}\sum_{s'\in\mathcal{S}_{k+1}}f^{(t,\tau)}(s,a,s')\theta_{t,\tau}^*(s,a,s')$$

$$\leq-\overline{\sigma}\left\|\frac{\mathbf{Q}(t)}{V}\right\|_2+\overline{\sigma}B+L,$$

where the first inequality is due to the weak error bound in Lemma 4.2 and weak duality with $\theta_{t,\tau}^*$ being one primal solution, the second inequality is by triangle inequality, and the third inequality is by Assumption 2.3 and Assumption 4.1. On the other hand, when $\mathrm{dist}(\frac{\mathbf{Q}(t)}{V},\mathcal{V}_{t,\tau}^*)\leq\overline{\vartheta}$, we have

$$q^{(t,\tau)}\left(\frac{\mathbf{Q}(t)}{V}\right)=\min_{\theta\in\Delta}\left\langle f^{(t,\tau)},\theta\right\rangle+\sum_{i=1}^{I}\frac{Q_i(t)}{V}(\langle g_i,\theta\rangle-c_i)$$

$$=\min_{\theta\in\Delta}\left\langle f^{(t,\tau)},\theta\right\rangle+\sum_{i=1}^{I}[\eta_{t,\tau}^*]_i(\langle g_i,\theta\rangle-c_i)+\sum_{i=1}^{I}\left(\frac{Q_i(t)}{V}-[\eta_{t,\tau}^*]_i\right)(\langle g_i,\theta\rangle-c_i)$$

$$\leq q^{(t,\tau)}(\eta_{t,\tau}^*)+\left\|\frac{\mathbf{Q}(t)}{V}-\eta_{t,\tau}^*\right\|_2\|\mathbf{g}(\theta)-\mathbf{c}\|_2$$

$$\leq L+2\overline{\vartheta}L,$$

where the first inequality is by the definition of $q^{(t,\tau)}(\eta_{t,\tau}^*)$ and Cauchy-Schwarz inequality, and the second inequality is due to weak duality and Assumption 2.3 such that

$$q^{(t,\tau)}(\eta_{t,\tau}^*)\leq\left\langle f^{(t,\tau)},\theta_{t,\tau}^*\right\rangle\leq\left\|f^{(t,\tau)}\right\|_\infty\|\theta_{t,\tau}^*\|_1\leq L,$$

$$\left\|\frac{\mathbf{Q}(t)}{V}-\eta_{t,\tau}^*\right\|_2\|\mathbf{g}(\theta)-\mathbf{c}\|_2\leq\overline{\vartheta}\sqrt{\sum_{i=1}^{I}\left|\langle g_i,\theta\rangle-c_i\right|^2}\leq\overline{\vartheta}\sum_{i=1}^{I}(\|g_i\|_\infty\|\theta\|_1+|c_i|)\leq 2\overline{\vartheta}L.$$

Now we can combine the two cases as follows

$$q^{(t,\tau)}\left(\frac{\mathbf{Q}(t)}{V}\right)\leq-\overline{\sigma}\left\|\frac{\mathbf{Q}(t)}{V}\right\|_2+\overline{\sigma}B+2L+2\overline{\vartheta}L+\overline{\sigma}\overline{\vartheta}. \tag{37}$$

The bound in (37) is due to

**(1)** When $\mathrm{dist}\big(\frac{\mathbf{Q}(t)}{V}, \mathcal{V}^*_{t,\tau}\big) \geq \overline{\vartheta}$, we have

$$q^{(t,\tau)}\big(\frac{\mathbf{Q}(t)}{V}\big) \leq -\overline{\sigma}\big\|\frac{\mathbf{Q}(t)}{V}\big\|_2 + \overline{\sigma}B + L \leq -\overline{\sigma}\big\|\frac{\mathbf{Q}(t)}{V}\big\|_2 + \overline{\sigma}B + 2L + 2\overline{\vartheta}L + \overline{\sigma}\overline{\vartheta}.$$

**(2)** When $\mathrm{dist}\big(\frac{\mathbf{Q}(t)}{V}, \mathcal{V}^*_{t,\tau}\big) < \overline{\vartheta}$, we have

$$q^{(t,\tau)}\big(\frac{\mathbf{Q}(t)}{V}\big) \leq L + 2\overline{\vartheta}L \leq -\overline{\sigma}\big\|\frac{\mathbf{Q}(t)}{V}\big\|_2 + \overline{\sigma}B + 2L + 2\overline{\vartheta}L + \overline{\sigma}\overline{\vartheta},$$

since $-\overline{\sigma}\big\|\frac{\mathbf{Q}(t)}{V}\big\|_2 + \overline{\sigma}\overline{\vartheta} + \overline{\sigma}B \geq -\overline{\sigma}\cdot\mathrm{dist}\big(\frac{\mathbf{Q}(t)}{V}, \mathcal{V}^*_{t,\tau}\big) + \overline{\sigma}\overline{\vartheta} + \overline{\sigma}B - \overline{\sigma}B = \overline{\sigma}\big[-\mathrm{dist}\big(\frac{\mathbf{Q}(t)}{V}, \mathcal{V}^*_{t,\tau}\big) + \overline{\vartheta}\big] \geq 0$.

Therefore, plugging (37) into (36), we can obtain that given the event $\mathcal{D}_T$ happens, the following holds

$$\begin{aligned}
\mathbb{E}[\|\mathbf{Q}(\tau+t)\|_2^2 &- \|\mathbf{Q}(t)\|_2^2 | \mathcal{F}^t, \mathcal{D}_T] \\
&\leq 2\tau^2 L + \tau C_{V,\alpha,\lambda} + 2\alpha L\log(|\mathcal{S}|^2|\mathcal{A}|/\lambda) - 2\tau\overline{\sigma}\|\mathbf{Q}(t)\|_2,
\end{aligned} \tag{38}$$

where we define

$$C_{V,\alpha,\lambda} := 2(\overline{\sigma}B + \overline{\sigma}\,\overline{\vartheta})V + (6 + 4\overline{\vartheta})VL + \frac{VL}{\alpha} + 4L\lambda V + 2\alpha\lambda L\log|\mathcal{S}|^2|\mathcal{A}| + 8L^2$$

We can see that if $\|\mathbf{Q}(t)\|_2 \geq (2\tau L + C_{V,\alpha,\lambda})/\overline{\sigma} + 2\alpha\lambda L\log(|\mathcal{S}|^2|\mathcal{A}|/\lambda)/(\overline{\sigma}\tau) + \tau\overline{\sigma}/2$, then according to (38), there is

$$\begin{aligned}
\mathbb{E}[\|\mathbf{Q}(\tau+t)\|^2 | \mathcal{F}^t, \mathcal{D}_T] &\leq \|\mathbf{Q}(t)\|^2 - \tau\overline{\sigma}\|\mathbf{Q}(t)\|_2 - \frac{\overline{\sigma}^2\tau^2}{2} \\
&\leq \|\mathbf{Q}(t)\|_2^2 - \tau\overline{\sigma}\|\mathbf{Q}(t)\|_2 + \frac{\overline{\sigma}^2\tau^2}{4} \\
&\leq \Big(\|\mathbf{Q}(t)\|_2 - \frac{\tau\overline{\sigma}}{2}\Big)^2.
\end{aligned}$$

Due to $\|\mathbf{Q}(t)\|_2 \geq \frac{\tau\overline{\sigma}}{2}$ and by Jensen's inequality, we have

$$\mathbb{E}[\|\mathbf{Q}(\tau+t)\|_2 | \mathcal{F}^t, \mathcal{D}_T] \leq \sqrt{\mathbb{E}[\|\mathbf{Q}(\tau+t)\|_2^2 | \mathcal{F}^t, \mathcal{D}_T]} \leq \|\mathbf{Q}(t)\|_2 - \frac{\tau\overline{\sigma}}{2}. \tag{39}$$

Then we can compute the expectation $\mathbb{E}[\|\mathbf{Q}(\tau+t)\|_2^2 - \|\mathbf{Q}(t)\|_2^2 | \mathcal{F}^t]$ according to the law of total expectation. With (33) and (39), we can obtain that

$$\begin{aligned}
\mathbb{E}[\|\mathbf{Q}(\tau+t)\|_2 &- \|\mathbf{Q}(t)\|_2 | \mathcal{F}^t] \\
&= P(\mathcal{D}_T)\mathbb{E}[\|\mathbf{Q}(\tau+t)\|_2 - \|\mathbf{Q}(t)\|_2 | \mathcal{F}^t, \mathcal{D}_T] + P(\overline{\mathcal{D}}_T)\mathbb{E}[\|\mathbf{Q}(\tau+t)\|_2 - \|\mathbf{Q}(t)\|_2 | \mathcal{F}^t, \overline{\mathcal{D}}_T] \\
&\leq -\frac{\tau\overline{\sigma}}{2}(1 - \zeta) + 2\zeta\tau L = -\tau\Big[\frac{\overline{\sigma}}{2} - \zeta\Big(\frac{\overline{\sigma}}{2} + 2L\Big)\Big] \leq -\frac{\overline{\sigma}}{4}\tau,
\end{aligned}$$

where we let $\overline{\sigma}/4 \geq \zeta(\overline{\sigma}/2 + 2L)$.

Summarizing the above results, we know that if $\overline{\sigma}/4 \geq \zeta(\overline{\sigma}/2 + 2L)$, then

$$|\|\mathbf{Q}(t+1)\|_2 - \|\mathbf{Q}(t)\|_2| \leq 2L,$$

$$\mathbb{E}[\|\mathbf{Q}(t+\tau)\|_2 - \|\mathbf{Q}(t)\|_2 | \mathcal{F}^t] \leq \begin{cases} 2\tau L, & \text{if } \|\mathbf{Q}(t)\|_2 < \psi \\ -\frac{\overline{\sigma}}{4}\tau, & \text{if } \|\mathbf{Q}(t)\|_2 \geq \psi \end{cases},$$

where we let

$$\psi = \frac{2\tau L + C_{V,\alpha,\lambda}}{\overline{\sigma}} + \frac{2\alpha L\log(|\mathcal{S}|^2|\mathcal{A}|/\lambda)}{\overline{\sigma}\tau} + \frac{\tau\overline{\sigma}}{2},$$

$$C_{V,\alpha,\lambda} = 2(\overline{\sigma}B + \overline{\sigma}\,\overline{\vartheta})V + (6 + 4\overline{\vartheta})VL + \frac{VL}{\alpha} + 4L\lambda V + 2\alpha\lambda L\log|\mathcal{S}|^2|\mathcal{A}| + 8L^2.$$

Directly by Lemma C.6, for a certain $t \in \{1, ..., T\}$, the following inequality holds with probability at least $1 - \delta$,

$$\|\mathbf{Q}(t)\|_2 \leq \psi + \tau \frac{512L^2}{\overline{\sigma}} \log \left( 1 + \frac{128L^2}{\overline{\sigma}^2} e^{\overline{\sigma}/(32L)} \right) + \tau \frac{64L^2}{\overline{\sigma}} \log \frac{1}{\delta}. \tag{40}$$

Further employing union bound for probabilities, we have that with probability at least $1 - T\delta$, for any $t \in \{1, \ldots, T\}$, the above inequality (40) holds.

We can understand the upper bound of the term $\log \left( 1 + \frac{128L^2}{\overline{\sigma}^2} e^{\overline{\sigma}/(32L)} \right)$ in the following way: **(1)** if $\frac{128L^2}{\overline{\sigma}^2} e^{\overline{\sigma}/(32L)} \geq 1$, then this term is bounded by $\log \left( \frac{256L^2}{\overline{\sigma}^2} e^{\overline{\sigma}/(32L)} \right) = \frac{\overline{\sigma}}{32L} + \log \frac{256L^2}{\overline{\sigma}^2}$; **(2)** if $\frac{128L^2}{\overline{\sigma}^2} e^{\overline{\sigma}/(32L)} < 1$, then the term is bounded by $\log 2$. Thus, we have

$$\log \left( 1 + \frac{128L^2}{\overline{\sigma}^2} e^{\overline{\sigma}/(32L)} \right) \leq \log 2 + \frac{\overline{\sigma}}{32L} + \log \frac{256L^2}{\overline{\sigma}^2}.$$

This discussion shows that the log term in (40) will not introduce extra dependency on $L$ except a $\log L$ term. This completes our proof. $\qquad \square$

## C.4  Proof of Lemma 5.5

**Lemma C.7** (Lemma 9 of Yu et al. [2017]). *Let $\{Z(t), t \geq 0\}$ be a supermartingale adapted to a filtration $\{\mathcal{F}^t, t \geq 0\}$ with $Z(0) = 0$ and $\mathcal{F}^0 = \{\emptyset, \Omega\}$, i.e., $\mathbb{E}[Z(t+1)|\mathcal{F}^t] \leq Z(t)$, $\forall t \geq 0$. Suppose there exists a constant $\varsigma > 0$ such that $\{|Z(t+1) - Z(t)| > \varsigma\} \subset \{Y(t) > 0\}$, where $Y(t)$ is process with $Y(t)$ adpated to $\mathcal{F}^t$ for all $t \geq 0$. Then, for all $z > 0$, we have*

$$Pr(Z(t) \geq z) \leq e^{-z^2/(2t\varsigma^2)} + \sum_{\tau=0}^{t-1} Pr(Y(\tau) > 0), \forall t \geq 1.$$

We are in position to give the proof of Lemma 5.5.

*Proof of Lemma 5.5.* Now we compute the upper bound of the term $\sum_{t=1}^{T} \sum_{i=1}^{I} Q_i(t-1)(\langle g_i^{t-1}, \overline{\theta}^* \rangle - c_i)$. Note that $Z(t) := \sum_{\tau=1}^{t} \sum_{i=1}^{I} Q_i(\tau-1)(\langle g_i^{\tau-1}, \overline{\theta}^* \rangle - c_i)$ is supermartigale which can be verified by

$$\mathbb{E}[Z(t)|\mathcal{F}^{t-1}] = \mathbb{E} \left[ \sum_{\tau=1}^{t} \sum_{i=1}^{I} Q_i(\tau-1)(\langle g_i^{\tau-1}, \overline{\theta}^* \rangle - c_i) \Big| \mathcal{F}^{t-1} \right]$$

$$= \sum_{i=1}^{I} \mathbb{E}[Q_i(t-1)|\mathcal{F}^{t-1}](\langle \mathbb{E}[g_i^{t-1}|\mathcal{F}^{t-1}], \overline{\theta}^* \rangle - c_i) + \sum_{\tau=1}^{t-1} \sum_{i=1}^{I} Q_i(\tau-1)(\langle g_i^{\tau-1}, \overline{\theta}^* \rangle - c_i)$$

$$\leq \sum_{\tau=1}^{t-1} \sum_{i=1}^{I} Q_i(\tau-1)(\langle g_i^{\tau-1}, \overline{\theta}^* \rangle - c_i) = \mathbb{E}[Z(t-1)],$$

where $Q_i(t-1)$ and $g_i^{t-1}$ are independent variables with $Q_i(t-1) \geq 0$ and $\langle \mathbb{E}[g_i^{t-1}|\mathcal{F}^{t-1}], \overline{\theta}^* \rangle \leq c_i$. On the other hand, we can know the random process has bounded drift as

$$|Z(t+1) - Z(t)| = \sum_{i=1}^{I} Q_i(t)(\langle g_i^t, \overline{\theta}^* \rangle - c_i)$$

$$\leq \|\mathbf{Q}(t)\|_2 \sqrt{\sum_{i=1}^{I} \left| \langle g_i^t, \overline{\theta}^* \rangle - c_i \right|^2}$$

$$\leq \|\mathbf{Q}(t)\|_2 \sum_{i=1}^{I} (\|g_i^t\|_\infty \|\overline{\theta}^*\|_1 + |c_i|) \leq 2L \|\mathbf{Q}(t)\|_2,$$

where the first inequality is by Cauchy-Schwarz inequality, and the last inequality is by Assumption 2.3. This also implies that for an arbitrary $\varsigma$, we have $\{|Z(t+1) - Z(t)| > \varsigma\} \subset \{Y(t) :=$

$\|\mathbf{Q}(t)\|_2 - \varsigma/(2L) > 0\}$ since $|Z(t+1) - Z(t)| > \varsigma$ implies $2L\|\mathbf{Q}(t)\|_2 > \varsigma$ according to the above inequality. Thus, by Lemma C.7, we have

$$Pr\left(\sum_{t=1}^{T}\sum_{i=1}^{I} Q_i(t-1)(\langle g_i^{t-1}, \overline{\theta}^*\rangle - c_i) \geq z\right) \leq e^{-z^2/(2T\varsigma^2)} + \sum_{t=0}^{T-1} Pr\left(\|\mathbf{Q}(t)\|_2 > \frac{\varsigma}{2L}\right), \quad (41)$$

where we could see that boundign $\|\mathbf{Q}(t)\|_2$ is the key to obtaining the bound of $\sum_{t=1}^{T}\sum_{i=1}^{I} Q_i(t-1)(\langle g_i^{t-1}, \overline{\theta}^*\rangle - c_i)$.

Next, we will show the upper bound of the term $\|\mathbf{Q}(t)\|_2$. According to Lemma 5.3, if $\overline{\sigma}/4 \geq \zeta(\overline{\sigma}/2 + 2L)$, setting

$$\psi = \frac{2\tau L + C_{V,\alpha,\lambda}}{\overline{\sigma}} + \frac{2\alpha L \log(|\mathcal{S}|^2|\mathcal{A}|/\lambda)}{\overline{\sigma}\tau} + \frac{\tau\overline{\sigma}}{2},$$

$$C_{V,\alpha,\lambda} := 2V\left(\overline{\sigma}B + 3L + 2\overline{\vartheta}L + \overline{\sigma}\overline{\vartheta} + \frac{L}{2\alpha} + 2L\lambda + \frac{\alpha\lambda L \log|\mathcal{S}|^2|\mathcal{A}| + 4L^2}{V}\right),$$

we have that with probability at least $1 - \delta$, for a certain $t \in \{1, \ldots, T\}$, the following inequality holds

$$\|\mathbf{Q}(t)\|_2 \leq \psi + \tau\frac{512L^2}{\overline{\sigma}}\log[1 + \frac{128L^2}{\overline{\sigma}^2}e^{\overline{\sigma}/(32L)}] + \tau\frac{64L^2}{\overline{\sigma}}\log\frac{1}{\delta}.$$

This inequality is equivalent to

$$Pr\left(\|\mathbf{Q}(t)\|_2 > \psi + \tau\frac{512L^2}{\overline{\sigma}}\log[1 + \frac{128L^2}{\overline{\sigma}^2}e^{\overline{\sigma}/(32L)}] + \tau\frac{64L^2}{\overline{\sigma}}\log\frac{1}{\delta}\right) \leq \delta.$$

Setting $\varsigma = 2L\psi + \tau\frac{1024L^3}{\overline{\sigma}}\log\left[1 + \frac{128L^2}{\overline{\sigma}^2}e^{\overline{\sigma}/(32L)}\right] + \tau\frac{128L^3}{\overline{\sigma}}\log\frac{1}{\delta}$ and $z = \sqrt{2T\varsigma^2\log\frac{1}{T\delta}}$ in (41), then the following probability hold with probability at least $1 - 2T\delta$ with

$$\sum_{t=0}^{T-1}\sum_{i=1}^{I} Q_i(t)(\langle g_i^t, \overline{\theta}^*\rangle - c_i)$$

$$\leq \left(2L\psi + \tau\frac{1024L^3}{\overline{\sigma}}\log\left[1 + \frac{128L^2}{\overline{\sigma}^2}e^{\overline{\sigma}/(32L)}\right] + \tau\frac{128L^3}{\overline{\sigma}}\log\frac{1}{\delta}\right)\sqrt{T\log\frac{1}{T\delta}},$$

which completes the proof. $\qquad\qquad\qquad\qquad\qquad\qquad\qquad\qquad\qquad\qquad\qquad\qquad\square$

# D  Proofs of Lemmas in Section 5.2

## D.1  Proof of Lemma 5.6

*Proof of Lemma 5.6.* We start our proof with the updating rule of $\mathbf{Q}(\cdot)$ as follows

$$Q_i(t) = \max\{Q_i(t-1) + \langle g_i^{t-1}, \theta^t\rangle - c_i, 0\}$$
$$\geq Q_i(t-1) + \langle g_i^{t-1}, \theta^t\rangle - c_i$$
$$\geq Q_i(t-1) + \langle g_i^{t-1}, \theta^{t-1}\rangle - c_i + \langle g_i^{t-1}, \theta^t - \theta^{t-1}\rangle.$$

Rearranging the terms in the above inequality futher leads to

$$\langle g_i^{t-1}, \theta^{t-1}\rangle - c_i \leq Q_i(t) - Q_i(t-1) - \langle g_i^{t-1}, \theta^t - \theta^{t-1}\rangle.$$

Thus, taking summation on both sides of the above inequality from $0$ to $T-1$ leads to

$$\sum_{t=0}^{T-1}(\langle g_i^t, \theta^t\rangle - c_i) \leq Q_i(T) - \sum_{t=0}^{T-1}\langle g_i^t, \theta^{t+1} - \theta^t\rangle$$

$$\leq Q_i(T) + \sum_{t=0}^{T-1}\|g_i^t\|_\infty\|\theta^{t+1} - \theta^t\|_1,$$

where the second inequality is due to Hölder's inequality. Note that the right-hand side of the above inequality is no less than 0 since $Q_i(t) = \max\{Q_i(t-1) + \langle g_i^{t-1}, \theta^t \rangle - c_i, 0\} \geq 0$. Thus, we have

$$\left[ \sum_{t=0}^{T-1} (\langle g_i^t, \theta^t \rangle - c_i) \right]_+ \leq Q_i(T) + \sum_{t=0}^{T-1} \|g_i^t\|_\infty \|\theta^{t+1} - \theta^t\|_1.$$

Defining $\mathbf{g}^t(\theta^t) := [\langle g_1^t, \theta^t \rangle, \cdots, \langle g_I^t, \theta^t \rangle]^\top$ and $\mathbf{c} := [c_1, \cdots, c_I]^\top$, we would obtain

$$\left\| \left[ \sum_{t=0}^{T-1} (\mathbf{g}^t(\theta^t) - \mathbf{c}) \right]_+ \right\|_2 \leq \|\mathbf{Q}(T)\|_2 + \sum_{t=0}^{T-1} \sqrt{\sum_{i=1}^{I} \|g_i^t\|_\infty^2} \|\theta^{t+1} - \theta^t\|_1$$

$$\leq \|\mathbf{Q}(T)\|_2 + \sum_{t=0}^{T-1} \sum_{i=1}^{I} \|g_i^t\|_\infty \|\theta^{t+1} - \theta^t\|_1$$

$$\leq \|\mathbf{Q}(T)\|_2 + \sum_{t=1}^{T} \|\theta^t - \theta^{t-1}\|_1,$$

where the third inequality is due to Assumption 2.3. This completes the proof. $\qquad\square$

## D.2   Proof of Lemma 5.7

**Lemma D.1** (Proposition 18 of Jaksch et al. [2010]). *The number of epochs up to episode $T$ with $T \geq |\mathcal{S}||\mathcal{A}|$ is upper bounded by*

$$\ell(T) \leq |\mathcal{S}||\mathcal{A}| \log\left( \frac{8T}{|\mathcal{S}||\mathcal{A}|} \right) \leq \sqrt{T|\mathcal{S}||\mathcal{A}|} \log\left( \frac{8T}{|\mathcal{S}||\mathcal{A}|} \right),$$

*where $\ell(\cdot)$ is a mapping from a certain episode to the epoch where it lives.*

We are ready to give the proof of Lemma 5.7.

*Proof of Lemma 5.7.* We need to discuss the upper bound of the term $\|\theta^t - \theta^{t-1}\|_1$ in two different cases:

**(1)** $\ell(t) = \ell(t-1)$, i.e., episodes $t$ and $t-1$ are in the same epoch;

**(2)** $\ell(t) > \ell(t-1)$, i.e., episodes $t$ and $t-1$ are in two different epochs.

For the first case where $\ell(t) = \ell(t-1)$, according to Lemma C.3, letting $\mathbf{x}^{opt} = \theta^t$, $\mathbf{y} = \widetilde{\theta}^{t-1}$, $\mathbf{z} = \widetilde{\theta}^{t-1}$ and $h(\theta) = \langle Vf^{t-1} + \sum_{i=1}^{I} Q_i(t-1)g_i^{t-1}, \theta \rangle$, we have

$$\left\langle Vf^{t-1} + \sum_{i=1}^{I} Q_i(t-1)g_i^{t-1}, \theta^t \right\rangle + \alpha D(\theta^t, \widetilde{\theta}^{t-1})$$

$$\leq \left\langle Vf^{t-1} + \sum_{i=1}^{I} Q_i(t-1)g_i^{t-1}, \widetilde{\theta}^{t-1} \right\rangle + \alpha D(\widetilde{\theta}^{t-1}, \widetilde{\theta}^{t-1}) - \alpha D(\widetilde{\theta}^{t-1}, \theta^t)$$

$$= \left\langle Vf^{t-1} + \sum_{i=1}^{I} Q_i(t-1)g_i^{t-1}, \widetilde{\theta}^{t-1} \right\rangle - \alpha D(\widetilde{\theta}^{t-1}, \theta^t).$$

Rearranging the terms and dropping the last term (due to $D(\theta^{t-1}, \theta^t) \geq 0$) yield

$$\alpha D(\theta^t, \widetilde{\theta}^{t-1}) \leq \left\langle Vf^{t-1} + \sum_{i=1}^{I} Q_i(t-1)g_i^{t-1}, \widetilde{\theta}^{t-1} - \theta^t \right\rangle + \alpha D(\theta^{t-1}, \widetilde{\theta}^{t-1})$$

$$\leq \left( V\|f^{t-1}\|_\infty + \sum_{i=1}^{I} Q_i(t-1)\|g_i^{t-1}\|_\infty \right) \|\widetilde{\theta}^{t-1} - \theta^t\|_1$$

$$\leq \left( V + \|\mathbf{Q}(t-1)\|_2 \sqrt{\sum_{i=1}^{I} \|g_i^{t-1}\|_\infty^2} \right) \|\widetilde{\theta}^{t-1} - \theta^t\|_1$$

$$\leq (V + \|\mathbf{Q}(t-1)\|_2)\|\widetilde{\theta}^{t-1} - \theta^t\|_1,$$

where the second inequality is by Hölder's inequality and triangle inequality, the third inequality is by Assumption 2.3, and the last inequality is due to Assumption 2.3. Note that by Lemma C.4, there is

$$D(\theta^t, \widetilde{\theta}^{t-1}) \geq \frac{1}{2L}\|\theta^t - \widetilde{\theta}^{t-1}\|_1^2.$$

Thus, combining the previous two inequalities, we obtain

$$\|\theta^t - \widetilde{\theta}^{t-1}\|_1^2 \leq \frac{2LV + 2L\|\mathbf{Q}(t-1)\|_2}{\alpha}\|\theta^t - \widetilde{\theta}^{t-1}\|_1.$$

The, we obtain the upper bound of $\|\theta^t - \widetilde{\theta}^{t-1}\|_1$ as follows

$$\|\theta^t - \widetilde{\theta}^{t-1}\|_1 \leq \frac{2LV + 2L\|\mathbf{Q}(t-1)\|_2}{\alpha}.$$

Since there is

$$\|\theta^t - \widetilde{\theta}^{t-1}\|_1 = \sum_{k=0}^{L-1} \left\|\theta_k^t - (1-\lambda)\theta_k^{t-1} - \lambda\frac{1}{|\mathcal{S}|^2|\mathcal{A}|}\mathbf{1}\right\|_1 \geq (1-\lambda)\|\theta^t - \theta^{t-1}\|_1 - \lambda L,$$

where $\theta_k := [\theta(s, a, s')]_{s \in \mathcal{S}_k, a \in \mathcal{A}, s' \in \mathcal{S}_{k+1}}$, we further have

$$\|\theta^t - \theta^{t-1}\|_1 \leq \frac{2LV + 2L\|\mathbf{Q}(t-1)\|_2}{(1-\lambda)\alpha} + \frac{\lambda L}{1-\lambda}. \tag{42}$$

For the second case where $\ell(t) > \ell(t-1)$, it is difficult to know whether the two solutions $\theta^{t-1}$ and $\theta^t$ are in the same feasible set since $\Delta(\ell(t)) \neq \Delta(\ell(t-1))$. Thus, the above derivation does not hold. Then, we give a bound for the term $\|\theta^t - \theta^{t-1}\|_1$ as follows

$$\|\theta^t - \theta^{t-1}\|_1 \leq \|\theta^t\|_1 + \|\theta^{t-1}\|_1 = \sum_{k=0}^{L-1}\sum_{s,a,s'} \theta^t(s,a,s') + \sum_{k=0}^{L-1}\sum_{s,a,s'} \theta^t(s,a,s') = 2L. \tag{43}$$

However, we can observe that $\ell(t) > \ell(t-1)$ only happens when $t$ is a starting episode for a new epoch, whose number in $T$ episodes is bounded by the number of epochs in $T$ episodes. According to Lemma D.1, the total number of epochs $\ell(T)$ is bounded by $\ell(T) \leq \sqrt{T|\mathcal{S}||\mathcal{A}|}\log[8T/(|\mathcal{S}||\mathcal{A}|)]$ which only grows in the order of $\log T$.

Thus, we can decompose the term $\sum_{t=1}^{T} \|\theta^t - \theta^{t-1}\|_1$ in the following way

$$\sum_{t=1}^{T} \|\theta^t - \theta^{t-1}\|_1 = \sum_{\substack{t:\, t \leq T,\\ \ell(t) > \ell(t-1)}} \|\theta^t - \theta^{t-1}\|_1 + \sum_{\substack{t:\, t \leq T,\\ \ell(t) = \ell(t-1)}} \|\theta^t - \theta^{t-1}\|_1$$

$$\leq 2L\ell(T) + \sum_{\substack{t:\, t \leq T,\\ \ell(t) = \ell(t-1)}} \|\theta^t - \theta^{t-1}\|_1,$$

where the inequality is due to (43) and the fact that $\sum_{\substack{t:\ t\leq T,\\ \ell(t)>\ell(t-1)}} 1 \leq \ell(T)$. By (42), we can further bound the last term in the above inequality as

$$\sum_{\substack{t:\ t\leq T,\\ \ell(t)=\ell(t-1)}} \|\theta^t - \theta^{t-1}\|_1$$

$$\leq \sum_{t=1}^{T} \left[\frac{2LV + 2L\|\mathbf{Q}(t-1)\|_2}{(1-\lambda)\alpha} + \frac{\lambda L}{1-\lambda}\right]$$

$$\leq \frac{2L}{(1-\lambda)\alpha} \sum_{t=0}^{T-1} \|\mathbf{Q}(t)\|_2 + \frac{2V + \alpha\lambda}{(1-\lambda)\alpha} LT.$$

This will eventually lead to

$$\sum_{t=1}^{T} \|\theta^t - \theta^{t-1}\|_1$$

$$\leq 2L\ell(T) + \sum_{\ell(t)=\ell(t-1)} \|\theta^t - \theta^{t-1}\|_1$$

$$\leq 2L\sqrt{T|\mathcal{S}||\mathcal{A}|} \log \frac{8T}{|\mathcal{S}||\mathcal{A}|} + \frac{2L}{(1-\lambda)\alpha} \sum_{t=0}^{T-1} \|\mathbf{Q}(t)\|_2 + \frac{2V + \alpha\lambda}{(1-\lambda)\alpha} LT,$$

which completes the proof. $\qquad\square$

## Footnotes

[2]We let $\operatorname{dist}(\mathbf{x}, \Lambda^*) := \min_{\mathbf{x}' \in \Lambda^*} \frac{1}{2}\|\mathbf{x} - \mathbf{x}'\|_2^2$ as the Euclidean distance between a point $\mathbf{x}$ and the set $\Lambda^*$.