[Reviews · NeurIPS 2020]

Review 1

Summary and Contributions: In this paper the authors focus on episodic reinforcement learning in constrained MDPs, where the optimized function may vary between episodes, and is only observed at the end of each episode (as are the stochastic constraints). This paper starts with a description of prior work which allows to position its theoretical contribution. Once the problem is formally described, the proposed algorithm (UCPD Mirror Descent) is presented. Finally, the bounds on regret as well as on the violation of constraints implied by the algorithm are demonstrated.

Strengths: This paper is very well written, the proofs it contains seems to be correct, and it makes an important theoretical contribution. The study of constrained MDPs is important because many systems have safety constraints to respect. To my knowledge this work is new, and of course very relevant to the NeurIPS community.

Weaknesses: My only concern with this paper is the lack of empirical evaluation of the algorithm, which makes it impossible to validate the use of this approach in practice. Some minor remarks are given in the "Correctness" and "Clarity" sections.

Correctness: This paper demonstrates quite clearly the existence of upper bounds on regret and constraint violations, based on many lemmas that are shown in the supplementary material (because of lack of space). The proofs in the paper seems to me to be correct. The only inconsistency seems to be between Equation 1 and Assumption 2.2: the sum is on {0,...,T-1} (resp. {1,...,T}) and with a factor 1/T (resp. without) for equation 1 (resp. assumption 2.2).

Clarity: The following remarks may help to improve the clarity of this paper. line 989: < constraint function > constraint function (also called budget functions) \overline{theta}^* should be defined in line 130 line 194: < Therefore we know that > Per definition Lemma 4.2: < for any x in X > for any eta in R^I line 226: < in our proof. . Then, this lemma > in our proof. Then, this lemma

Relation to Prior Work: The introduction is more than one page long and contains mostly elements on previous work related to the contributions of this paper.

Reproducibility: Yes

Additional Feedback: I thank the authors for their response.


Review 2

Summary and Contributions: This paper proposes an upper confidence primal-dual algorithm for CMDP where the losses could be adversarial and the model is unknown. The proposed algorithm is shown to achieve nearly optimal regret and a similar rate in the constraint violation. =======After rebuttal========= I have read the other reviews and the rebuttal. Given the results in [Yu et al. 2017], I tend to agree with reviewer #3. The two extensions mentioned in this paper, (1) from online gradient descent to mirror descent; (2) from knowing the transition model to introducing the \lambda-greedy exploration, don't seem to be hugely significant. On the other hand, combining various techniques to achieve a state-of-the-art regret bound in a more general setting is still an interesting result. Therefore, I lower my score to weak acceptance.

Strengths: The results on the paper extend the analysis in Rosenberg and Mansour (2019) to achieve a low constraint violation, by incorporating a constraint related dual multiplier into the objective function in solving the occupancy measure. The regret rate is nearly optimal.

Weaknesses: I only have minor comments. Technical intuition behind the dual multiplier Q is not explained in the main content of the paper. Also, could the authors comment on the importance of the loop-free assumption of the state space to the final results? (I understand this assumption is also made in Rosenberg and Mansour (2019))

Correctness: I have not checked the proofs in the appendix. The results seem correct to me.

Clarity: Overall the paper is well written and its trend is easy to follow.

Relation to Prior Work: Yes, this paper provides a good literature review.

Reproducibility: Yes

Additional Feedback: Typos: page 6, footnote: 'argmin' -> min Line 203: what is f^(t, \tau) (\theta)? <f^(t, \tau), \theta>? Same for g_i(\theta)


Review 3

Summary and Contributions: The authors consider the learning problem with episodic constraint Markov decision processes (episodic CMDPs). There are a total number of $I$ budget constraints that the learner must try to obey. The constraints are stochastic whose realizations the learner only observes at the end of each episode along with the loss function (full-information setting). The learner also observes the unknown Markov transition function by interacting with the environment (unknown dynamics). The authors proposed an algorithm that achieves both \thilde{O}(\sqrt{LS\sqrt{AT}}) regret and budget violations. The high-level approach follows that of “Online convex optimization with stochastic constraints” (Yu et al. NIPS’17), which considered a simpler stateless online learning setting. Both papers apply drift analysis on the quantity ||Q(t)||, to derive O(\sqrt{T}) bounds on ||Q(t)||, which leads to O(\sqrt{T}) bounds for regret and violations. This paper combines ideas from 1) online learning with stochastic constraints (Yu et al. 2017) and 2) episodic MDPs with unknown dynamics (Neu et al. 2012; Rosenberg et al. 2019), which has been studied as an online linear optimization problem. The paper also weakens previous working assumption (Slater condition) to Assumptions 4.1.

Strengths: The paper provides the first result for sublinear regret and violation bounds O(LS\sqrt{AT}) for constraint MDPs with unknown dynamics. Although the high-level approach follows Yu et al .2017 and Rosenberg et al 2019, the derivation of the bounds and the checking of the drift condition is more challenging. The paper provides a detailed derivation for its proofs. In regards to the proposed algorithm, based on the similar update rule for UC-O-REPS algorithm, it also enjoys a semi-closed form expression for the online mirror descent update at each time-step. The algorithm also adopts the doubling epoch length rule (as in Jaksch et al. 2010) to start new epochs.

Weaknesses: (W1): As such the high-level outline of the proof strategy follows previous procedures for drift analysis in (Yu et al. 2017) and MDP analysis in (Neu et al. 2012 and Rosenberg et al. 2019). Lemma B.2 is very similar to Lemma 4 in Neu et al. 2012 and Lemma B.2 in Rosenberg et al. 2019. Lemma 5.2 mirrors Lemma 8 in Yu et al. 2017. Technical lemmas for stochastic analysis are also from the previous paper: (Lemma B.6 and B.7 are Lemma 5 and 9 in Yu et al. 2017). The main lemma, Lemma 5.3, has the same goal as Lemma 7 in Yu et al. 2017, which is to show ||Q_t|| satisfies the drift condition stated in Lemma 5 in Yu et al. 2017. Lemma 5.6 is also exact as Lemma 3 in Yu et al. 2017. (In author feedback for W1): the authors replied that the goal of the paper is the theoretical study of the regret/budget violation bounds for episodic constrained MDPs. And indeed, they have designed algorithm 1 to obtain \tilde{O}(LS\sqrt{AT}) regret/budget violation bounds. (which matches the best upper bound for unconstrained episodic MDPs). Furthermore, in previous work, [Yu et al. 2017], the online gradient descent algorithm with 2-norm regularizer was studied, whereas in the current paper, the authors study the online mirror descent counterpart (with un-normalized KL divergence regularizer). Third, another contribution of the current paper, the authors argue, different from the stateless online convex optimization with stochastic constraints is the use of doubling epoch length, line 8 in Algorithm 1, (as used in UCRL2 [Jaksch et al. 2010]) to bound \sum_{t=1}^T || \theta^t - \theta^{t-1}||_1, in Lemma 5.7 (for bounding one of the terms that contributes to Term (IV)). However, the reviewer still finds great similarity. In short, by over-viewing the high-level regret decomposition in line 244, expression (9): the treatment for Term (II) is similar to that in [Yu et al. 2017] ; and the treatment for Term (I) can be found in [Rosenberg et al. 2019]. On the other hand, in line 263, expression (12) decomposes the budget violation bound into Term (III) and Term (IV). Term (IV) is treated similarly as in [Yu et al. 2017]. The differences in Term (II) and (IV) from previous work [Yu et al. 2017] mainly stem from the two papers using different regularizers. (2-norm or KL divergence) To give the authors credits, it still requires works to derive the bounds in Lemmas 5.2, 5.3 & 5.7. (W2): Regarding the algorithm, the hyper-parameters of the algorithm require the knowledge of the quantities (\hat{theta}, B and \sigma) specified in Assumption 4.1 and Lemma 4.2. It is not clear if these quantities can be accurately and easily measured in real applications. (In author feedback for W2): the only hyper-parameter involving the theoretical quantities (\hat{theta}, B and \sigma), is \zita the confidence parameter, which can be made much smaller with confidence interval increased only by a log scale.The quantities (\hat{theta}, B and \sigma) are for theoretical analysis only. The reviewer understand now the weak dependence of the hyper-parameters on the theoretical quantities (\hat{theta}, B and \sigma). Thus W2 is clear and is a non-issue. (W3): While providing an upper bound, there is no lower bound result; one cannot see if the upper bound dependence on other problem parameters (L, S, A) are tight. (In authors feedback for W3): under the setting of unknown MDP dynamics, the current regret lower bound for episodic MDP is \Omega(\sqrt{LSAT}), and the upper bound for episodic (unconstrained) MDPs is \tilde{O}(LS\sqrt{AT}). There is a gap of \sqrt{LS}. The current paper establishes regret and budget violation upper bounds for the episodic constrained MDPs of the order \tilde{O}(\sqrt{LSAT}), which matches the regret bound for the episodic unconstrained MDPs. It is left as future work to establish lower bound for constrained MDPs. (W4): (It is originally raised by reviewer #1.) In the paper, there are no numerical/empirical experiments.

Correctness: The high-level proof strategy is given in the main text and the proofs are written in detail in the appendix. The proofs are sound and provided in great details.

Clarity: The paper is written clearly with detailed technical proofs.

Relation to Prior Work: This paper’s high level approach is similar to that in Yu et al. 2017, with specialization for the MDP setting with further consideration of confidence bound analysis for unknown MDP dynamics, related to Neu et al. 2012 and Rosenberg et al. 2019. The statement that the weaker condition, Assumption 4.1, is implied by the Slater condition is proved by [Nedic & Ozdaglar 2009]. Lemma 4.2 is proved by Lemma 5 in [Wei et al. 2019]. These technical lemmas are proved by previous works.

Reproducibility: Yes

Additional Feedback: minor typos: line 542 and line 544, the term 4L^2/V should be (4L^2/V)T (and in the statement in Lemma 5.2 for this term, which does not have an effect on the regret bound) line 573, in expression (29), the superscripts for the summand should be 2\alpha [D(\theta, \tilde{\theta}^{t'}-D(\theta, \theta^{t'}) the same for line 575 line 611 should be \bar{\sigma}/4\geq \zeta(\bar{\sigma}/2 + 2L) line 402, should be "h(x) is a concave function" ... "all superlevel sets are closed and convex." line 485, missing logarithms for the un-normalized KL divergence. line 638, "boundign" line 646, on the RHS of the ineq., missing const. 2 inside the sqrt. line 668, "dropping the last term (due to D(\tilde{\theta}^{t-1}, \theta^t)\geq 0)" below line 668, on the RHS of the first ineq., it should be -\alpha D(\tilde{\theta}^{t-1}, \theta^t)


Review 4

Summary and Contributions: The authors addressed my concern in the rebuttal. I’d like to change the score to 7. This paper considers online episodic MDPs with constraints where the transition model is unknown and the loss function can change over time. The authors propose a new upper confidence primal-dual algorithm to solve the problem. Through theoretical analysis, authors show the UCB has sublinear regret and constraint violation when incorporating with primal-dual type approaches.

Strengths: Solving constrained MDP problems is a very promising topic. And the problem itself is very challenging given the fact that the reward function is time-dependent and the transition is unknown. The proposed algorithm achieves sqrt(T) regret and constraint violation which is almost the optimal result. Through rigorous analysis, the authors show this is nearly optimal.

Weaknesses: 1. Authors claim the proposed method outperforms OPDOP (Ding et al.,2020) given the fact that OPDOP’s result depends on |S, |A|, and L. However, I do not see that from your theoretical results. The regret and constraint violation bounds of the proposed algorithm are proportional to |S| sqrt(|A|) and L. 2. Even though Equ. (6) can be solved by an LP solver, there are still |S||A||S| decision variables. How do you solve such a large scale optimization in an efficient way? Could you present more details in terms of solving this optimization problem?

Correctness: Yes. The claimed theorem makes sense to me. And the proofs are correctly and technically novel.

Clarity: Yes. This paper is well written and easy to follow. Sufficient proofs are provided in order to understand the theoretical result.

Relation to Prior Work: Yes. Related works are well discussed and explained.

Reproducibility: Yes

Additional Feedback:

[Author Response · NeurIPS 2020]

We thank all reviewers for the valuable advice and questions. Our responses are provided below.

**Reviewer #1:** Thank you for your valuable suggestions. We are sorry for the typo causing confusions in Assumption 2.2. We will add the missing factor $1/T$ and change the summation to $\sum_{t=0}^{T-1}$. Furthermore, we will follow the suggestions to polish our paper and add the empirical comparisons.

**Reviewer #2: (Technical intuition behind the dual multiplier Q.)** At round $t$, we have obtained the constraints $\langle g_i^{t-1}, \theta \rangle - c_i \le 0$ for all $i \in [I]$. By Lagrangian duality theory, with these constraints and the loss function $\langle f^{t-1}, \theta \rangle$, we can have a Lagrange function whose dual variables are $Q_i$ for the $i$-th constraint for all $i \in [I]$ and should be non-negative. Then, the dual multiplier updating step in Eq. (5) can be viewed as a one-step dual ascent in an online setting. The operation $\max\{\cdot, 0\}$ is to guarantee the dual multiplier always non-negative. This is the main intuition behind the dual multiplier updating step. We will add this discussion in our paper.

**(Importance of the loop-free assumption.)** In our paper, this loop-free assumption is a standard assumption for episodic MDP. Technically, this assumption is a important for Lemma 5.1 to hold. Lemma 5.1 essentially gives the upper bound of the distance between the chosen occupancy measure and the true occupancy measure. Without the loop-free assumption, we need to develop new techniques to bound such distance. The loop-free assumption can potentially be weakened to the assumption that the underlying MDP has a fixed renewal state under any policy. We leave this as our future work.

**(Typo.)** In line 203, $f^{(t,\tau)}(\theta)$ and $g_i(\theta)$ should be corrected as $\langle f^{(t,\tau)}, \theta \rangle$ and $\langle g_i, \theta \rangle$. Thanks for pointing out the typo.

**Reviewer #3: (Techniques from existing works.)** We remark that the goal of this paper is to provide theoretical analysis of the constrained MDP scenario which are not fully studied before and present new bounds for this problem. In general, our paper provides a novel high probability bound for the mirror descent algorithms which is not a trivial extension of the paper Yu et al. 2017, as their paper only studied the online gradient algorithm in Euclidean space. On the other hand, our paper studies the problem without knowing the transition model, and involves the exploration step to deal with this challenge. Thus, we cannot directly apply Yu et al. 2017. Moreover, our work is beyond the simple constrained online learning setting and focuses on a more challenging constrained MDP problem. On the other hand, comparing to the previous work Rosenberg et al. 2019, we fully exploit the doubling of epoch length to obtain a sublinear constraint violation bound. This also provides a new insight on the application of epoch length doubling.

**(The hyper-parameters of the algorithm require the knowledge of $\overline{\vartheta}$, $B$, and $\overline{\sigma}$.)** As shown in Theorem 4.3, the settings of hyper-parameters $\alpha, V, \lambda$ in our algorithm do not need $\overline{\vartheta}$, $B$, and $\overline{\sigma}$. The constants $\overline{\vartheta}$, $B$, and $\overline{\sigma}$ are mainly for the purpose of theoretical analysis. Here $\overline{\sigma}$ is the only constant that are associated with a hyper-parameter $\zeta$, namely, $\zeta \in (0, 1/(4 + 8L/\overline{\sigma})]$. And $\zeta$ corresponds to the confidence interval $\varepsilon_\ell^\zeta$ defined as Eq. (3). In practice, we can set $\zeta$ sufficiently small, which will further guarantee that the probability $1 - 4\zeta$ in Theorem 4.3 is large. This will not affect the value of $\varepsilon_\ell^\zeta$ too much as it only depends on a factor of $\log^{1/2}(1/\zeta)$.

**(Is the dependence on parameters $L, S, A$ tight?)** As discussed in Jaksch et al., 2010, the lower bound of the regret for learning the *unconstrained* episodic MDP is $\Omega(\sqrt{L|\mathcal{S}||\mathcal{A}|T})$. The best known upper bound of the regret for the *unconstrained* episodic MDP is $\widetilde{\mathcal{O}}(L|\mathcal{S}|\sqrt{|\mathcal{A}|T})$ (Rosenberg and Mansour, 2019a) with a gap $\widetilde{\mathcal{O}}(\sqrt{L|\mathcal{S}|})$ to the lower bound. Different from the aforementioned works, in this paper, we study a constrained MDP. Intuitively, solving the constrained problem is more challenging than solving the unconstrained problem, as the class of the unconstrained MDPs is a subset of the constrained MDPs (since the constrained problem can be reduced to the unconstrained problem when the feasible set is the whole space.). For the constrained MDP, our paper can still obtain an $\widetilde{\mathcal{O}}(L|\mathcal{S}|\sqrt{|\mathcal{A}|T})$ regret, which matches the best known upper bound for the unconstrained MDP. But whether this is an optimal result remains to be explored. We leave the rigorous proof of the lower bound for the constrained MDP as our future work.

**Reviewer #4: (Comparisons to the result in Ding et al.,2020.)** In order to compare with the result in Ding et al.,2020, we introduce a new notation $|X|$ to denote the upper bound of the number of states at each layer. Thus, $|\mathcal{S}|$ in our paper is upper bounded by $L|X|$. Then, our regret bound and constraint violation are equivalently $\widetilde{\mathcal{O}}(\sqrt{L^4|X|^2|\mathcal{A}|T})$. In Ding et al.,2020, for the tabular case, the dimension $d$ is $|X||\mathcal{A}|$, $H$ is equivalently $L$, and $K$ is equivalent to $T$ in our paper. Thus, their results on regret bound and constraint violation can be rewritten as $\widetilde{\mathcal{O}}(\sqrt{L^8|X|^3|\mathcal{A}|^3T})$, which has worse dependencies on the factors $L, |X|, |\mathcal{A}|$.

**(More details of solving the constrained sub-problem.)** Solving the constrained sub-problem is basically in two steps: (1) perform an *unconstrained* mirror descent step, which admits a closed-form solution (Rosenberg and Mansour, 2019a); (2) project the iterate in the last step to the feasible set formed by the constraints. Note that the projection step 2 is another constrained minimization problem, whose objective is a KL divergence. Furthermore, we can reformulate this latter constrained minimization into its dual form, which is a convex optimization with *only non-negativity constraints*. Now this new problem can be efficiently solved as the constraints are much simpler than before. This algorithm is described in detail in Section 4.2 of Rosenberg and Mansour, 2019a. We will add this algorithm to our paper.

[Meta-Review · NeurIPS 2020]

I want to thank the authors for preparing the detailed rebuttal. This paper was discussed among all the reviewers during the post-rebuttal discussion phase. Overall, the reviewers are excited about this work on solving constrained MDP problems and have a positive assessment of the paper. All the reviewers acknowledged the theoretical contributions, especially in a challenging setting with unknown dynamics and non-stationary loss function. There was a clear consensus that the paper should be accepted. The reviewers have provided detailed feedback in their reviews, and we hope that the authors can incorporate this feedback when preparing the final version of the paper.